# SonicMoE: Accelerating MoE with IO and Tile-aware Optimizations

**Wentao Guo[1], Mayank Mishra[2], Xinle Cheng[1], Ion Stoica[2], Tri Dao[1,3]**
[1]Princeton University, [2]University of California, Berkeley, [3]Together AI
wg0420@princeton.edu, tri@tridao.me

## Abstract

Mixture of Experts (MoE) models have emerged as the de facto architecture for scaling up language models without significantly increasing the computational cost. Recent MoE models demonstrate a clear trend towards high expert granularity (smaller expert intermediate dimension) and higher sparsity (constant number of activated experts with higher number of total experts), which improves model quality per FLOP. However, fine-grained MoEs suffer from increased activation memory footprint and reduced hardware efficiency due to higher IO costs, while sparser MoEs suffer from wasted computations due to padding in Grouped GEMM kernels. In response, we propose a memory-efficient algorithm to compute the forward and backward passes of MoEs with minimal activation caching for the backward pass. We also design GPU kernels that overlap memory IO with computation benefiting all MoE architectures. Finally, we propose a novel "token rounding" method that minimizes the wasted compute due to padding in Grouped GEMM kernels. **As a result, our method SonicMoE reduces activation memory by 45% and achieves a 1.86x compute throughput improvement on Hopper GPUs compared to ScatterMoE's BF16 MoE kernel for a fine-grained 7B MoE.** Concretely, SonicMoE on 64 H100s achieves a training throughput of 213 billion tokens per day comparable to ScatterMoE's 225 billion tokens per day on 96 H100s for a 7B MoE model training with FSDP-2 using the lm-engine codebase. **On Blackwell GPUs, SonicMoE also achieves a 28.7% and 22.1% relative speedup on the forward and backward pass respectively compared to a highly optimized DeepGEMM baseline on OLMoE-sized 7B MoE models.** Under high MoE sparsity settings, our tile-aware token rounding algorithm yields an additional **1.16x** speedup on kernel execution time compared to vanilla top-$K$ routing while maintaining similar downstream performance. We open-source all our kernels[1] to enable faster MoE model training. An extended version of this paper can be found on arXiv[2].

## 1 Introduction

Mixture of Experts (MoE) (Shazeer et al., 2017) models have emerged as a key technique for scaling up parameters (Zhao et al., 2025a; Kimi et al., 2025) without increasing the training computational requirements. Modern transformers often have layers comprised of a sequence mixer block (e.g. Multi-head Attention (Vaswani et al., 2017)), followed by a channel mixer block (e.g. dense MLPs) where MoEs are an excellent substitute for dense MLPs for FLOPs efficiency. A MoE block is typically composed of a token router and multiple smaller and often equal-sized subnetworks, called "experts". MoEs can reduce FLOPs consumption during training by only activating a subset of all experts per token. However, reducing FLOPs does not directly translate to better hardware utilization since MoE computation features more dynamic IO accesses when each expert needs to gather token embeddings from different positions, and also scatter the results back to the original positions. Moreover, such hardware-unfriendliness becomes worse as experts become more *granular* (experts have smaller intermediate sizes) and *sparser* (experts are increased while keeping the number of activated experts constant), shown in Table 3.

---

[1] https://github.com/Dao-AILab/sonic-moe
[2] https://arxiv.org/pdf/2512.14080

MoE scaling laws (Clark et al., 2022; Krajewski et al., 2024) predict better model quality per FLOP with increasing expert granularity (ratio between the model's embedding dimension and each expert's intermediate size) and sparsity. Recent MoE models like DeepSeek V3 (DeepSeek-AI et al., 2024), Qwen3 MoE (QwenLM, 2025) and gpt-oss-120b (OpenAI, 2025), have demonstrated superior performance of "fine-grained" MoEs over "coarse-grained" MoEs at scale. Besides granularity, the pursuit of MoEs with better model quality while keeping computational requirements constant has also led to modern MoEs becoming sparser. For example, Kimi K2 (Kimi et al., 2025) has the same amount of activated parameters as DeepSeek V3 (DeepSeek-AI et al., 2024) but much larger total parameter count. Overall, granularity and sparsity for MoEs have only increased over time as shown in Table 3. We also note that the pursuit of granularity and sparsity is also adopted by recent alternative architectures to MoE such as PEER (He, 2024), Memory Layers (Berges et al., 2024), and Ultra-Mem (Huang et al., 2025).

Though more granular and sparser MoEs increase model quality per FLOP, they suffer from hardware inefficiency due to: (1) larger activation memory footprint for granular MoE models as activation size typically scales linearly with the number of activated experts, (2) lower arithmetic intensity and increased IO cost due to granular experts and (3) wasted computations due to tile quantization effects of grouped GEMM for highly sparse MoEs. The high granularity and sparsity both push MoE training towards the memory-bound regime requiring carefully designed MoE kernels to hide the increased IO costs. Existing state-of-the-art MoE kernels such as ScatterMoE (Tan et al., 2024) and MoMoE (Costin et al., 2025) are not designed to handle these high IO costs and they suffer significant training throughput degradation.

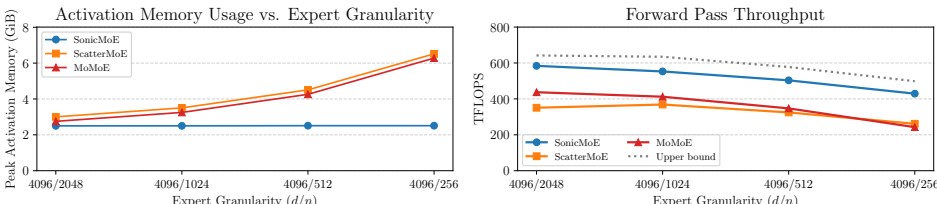

Figure 1: SonicMoE's per-layer activation memory footprint (left) stays constant even when expert granularity ($d/n$ where $d$ is the embedding dimension and $n$ is the expert intermediate dimension) increases, and is 0.20-1.59x more memory-efficient than other baselines. SonicMoE's forward computation throughput (right) reaches an average of 88% (max 91%, min 86%) of the upper bound (cuBLAS BMM + activation + cuBLAS BMM + aggregation) on H100 GPUs. Here we use a 30B MoE configuration with microbatch size of 32768 tokens, and we vary the activated experts / total number of experts as 2/32, 4/64, 8/128, and 16/256 from left to right.

**Summary of contributions.** We propose SonicMoE, a hardware and model architecture co-design solution to address MoE training efficiency problems, making the following contributions:

- **MoE training with minimum possible activation memory footprint without increasing FLOPs:** We analyze the impact of MoE granularity on the MoE layer's forward and backward passes and observe that increasing MoE granularity while maintaining constant FLOPs leads to a linear increase in activation memory required by the backward pass. Leveraging this observation, we carefully redesign the computation graph to avoid caching the activations for the router gradient computation while maintaining the mathematical equivalence to the original MoE formulation. As a result, for a fine-grained 7B MoE, SonicMoE reduces activation memory usage per layer by up to 45%.

- **Efficient MoE kernel that overlaps IO with computation to yield SOTA training throughput:** We show that increasing both granularity and sparsity leads to MoEs becoming increasingly memory bandwidth bound. To alleviate this bottleneck, we exploit the asynchrony of the GEMM and IO operations by overlapping them to maximize throughput. For the same fine-grained 7B MoE model, our approach increases relative speedup by 43% on the forward pass compared to a highly optimized DeepGEMM baseline on H100 GPUs, and by 83% and 115% on the backward pass compared to the state-of-the-art MoE baselines ScatterMoE and MoMoE, respectively. On B300 GPUs, our approach also achieves a 28.7% relative speedup on the forward pass and 22.1% on the backward pass compared to a highly optimized DeepGEMM baseline on OLMoE-sized 7B MoE models.

- **Token rounding routing that eliminates wasted FLOPs from sparse MoEs:** We introduce a drop-in routing algorithm that rounds the per-expert token counts to multiples of the tile size (e.g., 128) used by grouped GEMM in MoE kernels. This rounding reduces compute wasted on padding

while preserving the original token-to-expert assignment as much as possible. The algorithm ensures that, for each expert, the maximum deviation from the original top-$K$ token-choice result is bounded by one tile. This method effectively eliminates padding waste in grouped GEMM while maintaining the same total number of tokens in expectation, and it delivers robust token-choice accuracy even under highly sparse MoE training regimes. We validate the performance of this token-rounding strategy in a 1.4B-parameter sparse training setting, demonstrating that its compute throughput consistently exceeds that of the vanilla top-$K$ token-choice routing. In highly sparse regimes, the improvement reaches up to 16% higher TFLOPS for end-to-end MoE computation.

We release SonicMoE[1], mainly written in CuTe-DSL (NVIDIA, 2025a) with a PyTorch interface and a permissive license to benefit researchers and practitioners.

## 2 BACKGROUND

We first provide an overview of the MoE architecture and a standard MoE kernel employing grouped GEMM in Section 2.1. In Section 2.2, we discuss how granularity and MoE sparsity will affect MoE's training efficiency. We then examine the impact of MoE routing methods on the MoE model quality and training efficiency in Section 2.3.

### 2.1 MoE USING GROUPED GEMM

Modern GPUs support Tensor Cores; specialized hardware units with high matrix multiplication throughput (NVIDIA, 2022). A GEMM (general matrix multiply) (Lawson et al., 1979) kernel often has 3 stages: prologue (start input loading), mainloop (keep loading inputs and compute GEMM) and epilogue (miscellaneous IO/math operations on GEMM outputs). The kernel tiles computations (dividing large matrices into small tiles), and optionally pads dimensions so computation aligns with hardware-permissible tile sizes. In this paper, we follow standard GEMM notations in most BLAS (Lawson et al., 1979) libraries: we have $A \in \mathbb{R}^{\mathbf{M} \times \mathbf{K}}$, $B \in \mathbb{R}^{\mathbf{K} \times \mathbf{N}}$, $C \in \mathbb{R}^{\mathbf{M} \times \mathbf{N}}$ for $C = AB$ with problem shape $(\mathbf{M}, \mathbf{N}, \mathbf{K})$. This notation is adopted by CUTLASS (NVIDIA, 2025c) which implements efficient GEMM on CUDA.

---

**Algorithm 1** MoE forward with Grouped GEMM

**Input** : $X \in \mathbb{R}^{T \times d}$, $W_1 = \{W_{1,e}\}_{e \in [E]} \in \mathbb{R}^{d \times 2n}$, $W_2 = \{W_{2,e}\}_{e \in [E]} \in \mathbb{R}^{n \times d}$, routing scores $S \in \mathbb{R}^{T \times E}$, $\pi \in \{0,1\}^{T \times E}$ as a binary-valued mask matrix where $\pi_{t,e}$ represents whether token $t$ is routed to expert $e$.

**Output** : output activation $O \in \mathbb{R}^{T \times d}$

**Parallel for** $e \in [E]$ **do**
  $X_e \leftarrow \text{Gather}(X, \pi_{:,e})$
  // up-proj, varlen-$\mathbf{M}$ Grouped GEMM
  $H_e \leftarrow X_e W_{1,e}$
  $A_e \leftarrow \text{act\_func}(H_e)$
  // down-proj, varlen-$\mathbf{M}$ Grouped GEMM
  $Y_e \leftarrow A_e W_{2,e}$

**Parallel for** $t \in [T]$ **do**
  // expert aggregation
  $O_t = \sum_{e \in [E]} \pi_{t,e} S_{t,e} Y_{e,t}$

---

A MoE block is typically composed of a token router and multiple smaller and often equal-sized subnetworks, called "experts". The router is responsible for dispatching tokens to the experts. The outputs from all experts in the layer are then aggregated and passed onto the next layer. MoE computation can be performed using Grouped GEMM (a list of GEMMs with possibly different $\{\mathbf{M},\mathbf{N},\mathbf{K}\}$ dimensions). Algorithm 1 illustrates running MoE forward with Grouped GEMM. Here we refer to a Grouped GEMM operation with fixed $(\mathbf{N},\mathbf{K})$ dim but variable $\mathbf{M}$ as "varlen-$\mathbf{M}$ Grouped GEMM", while variable $\mathbf{K}$ but fixed $\mathbf{M}$ and $\mathbf{N}$ as "varlen-$\mathbf{K}$ Grouped GEMM".

### 2.2 MoE COMPUTATION

Arithmetic intensity, defined as the ratio of FLOPs over the number of transferred bytes (IO), is a metric to quantify whether a kernel is memory-bound (kernel runtime dominated by memory IO cost) or compute-bound (kernel runtime dominated by compute throughput).

The standard MoE computation for an expert $e$ with SwiGLU activation can be broken down into the following components:

$$H_e = \text{up-projection}(X_e) = X_e W_{1,e} : \mathbb{R}^{T_e \times d} \rightarrow \mathbb{R}^{T_e \times 2n} \qquad (1)$$

$$A_e = \text{SwiGLU}(H_e) : \mathbb{R}^{T_e \times 2n} \rightarrow \mathbb{R}^{T_e \times n} \qquad (2)$$

$$Y_e = \text{down-projection}(A_e) = A_e W_{2,e} : \mathbb{R}^{T_e \times n} \rightarrow \mathbb{R}^{T_e \times d} \qquad (3)$$

Here, the up-projection uses $4T_e nd$ FLOPs and $2T_e d + 4nd + 2T_e n$ HBM memory transfer bytes (we ignore the writes for $H_e$ here). Similarly, down-projection uses $2T_e nd$ FLOPs with $2T_e n + 2nd + 2T_e d$ bytes. Assuming $\rho = \frac{K}{E}$ as the MoE activation ratio, $G = \frac{d}{n}$ as the granularity and uniform routing i.e

$T_e = T\rho$, the arithmetic intensity (ignoring the writes for $H_e$) for the forward pass of an expert is

$$\frac{4T_e nd + 2T_e nd}{4T_e n + 6nd + 4T_e d} = \frac{3}{\frac{2}{d} + \frac{2}{n} + \frac{3}{T_e}} = \frac{3}{\frac{2+2G}{d} + \frac{3}{T\rho}} \tag{4}$$

For a specific model size (constant $d$), it can be seen that increasing granularity (increasing $G$) or increasing sparsity (decreasing $\rho$) leads to a decreasing arithmetic intensity. This is caused by the linear scaling of IO cost w.r.t. expert granularity. Therefore for the case of fine-grained MoEs (high $G$)[3], it becomes increasingly important to address the increased IO cost by maximally reducing IO access and hiding IO latency. We examine a memory-efficient MoE kernel design in Section 3 and discuss techniques to reduce IO access and latency in Section 4.

### 2.3 MoE ROUTING METHODS

In MoE, routing determines which experts to activate for each token. Token choice (TC) routing where each token independently selects the activated expert is often the default routing method for MoE models (Shazeer et al., 2017). We often have top-$K$ TC routing where the routing decision for token $t$ is $\text{TopK}_{e \in [E]}(S_{t,e}, K)$ and $S_{t,e}$ is the expert score for token $t$. Besides TC routing, expert choice (EC) routing is developed to avoid load imbalance for expert parallelism (Zhou et al., 2022) by letting experts choose the tokens. However, EC routing is not directly usable for inference because it is incompatible with autoregressive decoding, and switching back to TC at inference time leads to a mismatch. In addition, EC breaks causality by future token information leakage (Wang et al., 2024). To address the inference issue of EC routing, Raposo et al. (2024) introduce an auxiliary loss to promote the agreement between TC and EC routing results, or train an auxiliary router to explicitly predict the routing result of EC router and use this auxiliary router during inference.

In this paper, we propose a novel Grouped GEMM tile-aware token rounding method that rounds the number of received tokens per expert ("expert frequency") to nearby multiples of Grouped GEMM tile sizes and alters at most one tile of tokens per expert. This approach effectively reduces wasted FLOPs caused by Grouped GEMM padding during sparse MoE training while preserving inference quality of trained MoE models. There are similar works that propose to drop and reroute tokens, including Rectify-Router (Zeng et al., 2024), but they do not focus on the tile structure of Grouped GEMM. Other works such as TMA-adaptive FP8 Grouped GEMM (Fu et al., 2025) focus on reducing padding-related load traffic but the FLOPs wasted by non-aligned tile size in GEMM computation is not addressed.

## 3 MEMORY-EFFICIENT MoE ALGORITHM

The MoE computation in SonicMoE launches 8 kernels: during forward, we have up-proj ($A$), down-proj ($Y$), and expert aggregation ($O$) kernels; during backward, we have activation gradient kernels for $dH$ (down-proj), $d\tilde{X}$ (up-proj), $dX$ (aggregating $d\tilde{X}$), and weight gradient kernels $dW_1$ and $dW_2$. The overall computation is shown in Algorithm 2, 3, and 5, and the workflow is illustrated in Figure 8.

**Activation memory efficiency** We now focus on the key algorithmic objective: minimizing activation memory during training. The FLOPs of MoE forward and backward computation is $(6+12)TnKd$. For a given $T, d$, we need to keep $nK$ constant for constant FLOPs. Therefore, increasing granularity requires decreasing $n$ and proportionally increasing $K$. Hence, any activations with memory $O(TKd)$ should not be cached for backward computation to avoid activation memory scaling with granularity. For current MoE kernels like ScatterMoE, activations scale linearly with expert granularity.

Activations $Y$ (down-proj output) and $X_e$ (gathered $X$) have size $TKd$ and avoiding caching them eliminates activation memory dependency on granularity. We avoid writing $dY$ and $dO_e$ (gathered $dO$) to HBM as they increase the peak activation memory during the backward computation:

- For $X$ and $dO$, fusing the gather operation with the HBM load eliminates the need for materialization and activation caching in HBM.
- We identify an alternative computation path that does not involve using $Y$ and $dY$ to compute $dS$ and $dH$ without increasing FLOPs, as illustrated in Appendix D. This eliminates the need for caching $Y$.

---

[3]Here we refer to "fine-grained MoE" as MoE with large granularity i.e., $d$ is larger than $n$. We assume the setting of both iso-FLOPs and iso-params.

**Algorithm 2** SonicMoE's MoE kernel forward pass. Variables stored in HBM are colored blue. load and store means load from / store into HBM respectively.

**Input** : $X, S, \pi, W_1$, and $W_2$, same as Algorithm 1.
**Output :** MoE layer output $O$
Up-proj $A$ kernel $(X, W_1, \pi) \rightarrow (H, A)$:
// Gather + varlen-$\mathbf{M}$ Grouped GEMM + SwiGLU
    **Parallel for** $e \in [E]$ **do**
        $X_e, W_{1,e}, \pi_{:,e} \leftarrow \text{load}(X_e, W_{1,e}, \pi_{:,e})$
        $X_e \leftarrow \text{Gather}(X, \pi_{:,e})$
        $H_e \leftarrow X_e W_{1,e}$
        $A_e \leftarrow \text{act\_func}(H_e)$
        $H_e, A_e \leftarrow \text{store}(H_e, A_e)$
Down-proj $Y$ kernel $(A, W_2) \rightarrow Y$:
// varlen-$\mathbf{M}$ Grouped GEMM
    **Parallel for** $e \in [E]$ **do**
        $A_e, W_{2,e} \leftarrow \text{load}(A_e, W_{2,e})$
        $Y_e \leftarrow A_e W_{2,e}$
        $Y_e \leftarrow \text{store}(Y_e)$
Expert aggregation $O$ Kernel $(Y, S, \pi) \rightarrow O$:
// Gather and sum
    **Parallel for** $t \in [T]$ **do**
        $Y_{e,t}, S_{t,e}, \pi_{t,e} \leftarrow \text{load}(Y_{e,t}, S_{t,e}, \pi_{t,e})$
        $O_t \leftarrow \sum_{e \in [E]} \pi_{t,e} S_{t,e} Y_{e,t}$
        $O_t \leftarrow \text{store}(O_t)$

**Algorithm 3** SonicMoE's MoE kernel backward pass of down projection.

**Input** : $S, \pi, W_2, dO$.
**Output :** $dH, dW_2, dS$.
Down-proj act $dH$ kernel $(dO, W_2, S, \pi) \rightarrow (dH, dS, A')$:
// Gather + varlen-$\mathbf{M}$ Grouped GEMM + dSwiGLU + $dS$
// Appendix D elaborates this algorithm in more detail
    **Parallel for** $e \in [E]$ **do**
        $dO_e, W_{2,e}, S, \pi_{:,e} \leftarrow \text{load}(dO_e, W_{2,e}, S, \pi_{:,e})$
        $dO_e \leftarrow \text{Gather}(dO, \pi_{:,e})$
        // $dA'$ is a temp variable for computing $dA, dS$, and $A'$
        $dA'_e \leftarrow dO_e W_{2,e}^\top$            // $dA'_e \in \mathbb{R}^{T_e \times n}$
        $\mathbf{s}_e \leftarrow \text{Gather}(S, \pi_{:,e})$
        $dA_e \leftarrow \text{Broadcast}(\mathbf{s}_e) \, dA'_e$
        $A_e, dH_e \leftarrow \text{dAct\_func}(dA_e, H_e)$
        $A'_e \leftarrow \text{Broadcast}(\mathbf{s}_e) \, A_e$      // input for $dW_2$
        $dS_{e,t} \leftarrow \langle dA'_{e,t}, A_{e,t} \rangle$      // reduce over $n$ dim
        $dH_e, dS, A'_e \leftarrow \text{store}(dH_e, dS, A'_e)$
Down-proj weight $dW_2$ kernel $(dO, A', \pi) \rightarrow dW_2$:
// Gather + varlen-$\mathbf{K}$ Grouped GEMM
    **Parallel for** $e \in [E]$ **do**
        $dO_e, A'_e, \pi_{:,e} \leftarrow \text{load}(dO_e, A'_e, \pi_{:,e})$
        $dO_e \leftarrow \text{Gather}(dO, \pi_{:,e})$
        $dW_{2,e} \leftarrow A'^\top_e dO_e^\top$
        $dW_{2,e} \leftarrow \text{store}(dW_{2,e})$

As a result, we only cache $X, H$, and routing metadata for a total size about $2Td + 4TKn$ bytes per layer. **This activation memory usage is the minimum required for backward computation without doing activation recomputation with GEMM, and is independent of expert granularity.**

## 4 IO-AWARE KERNEL DESIGN

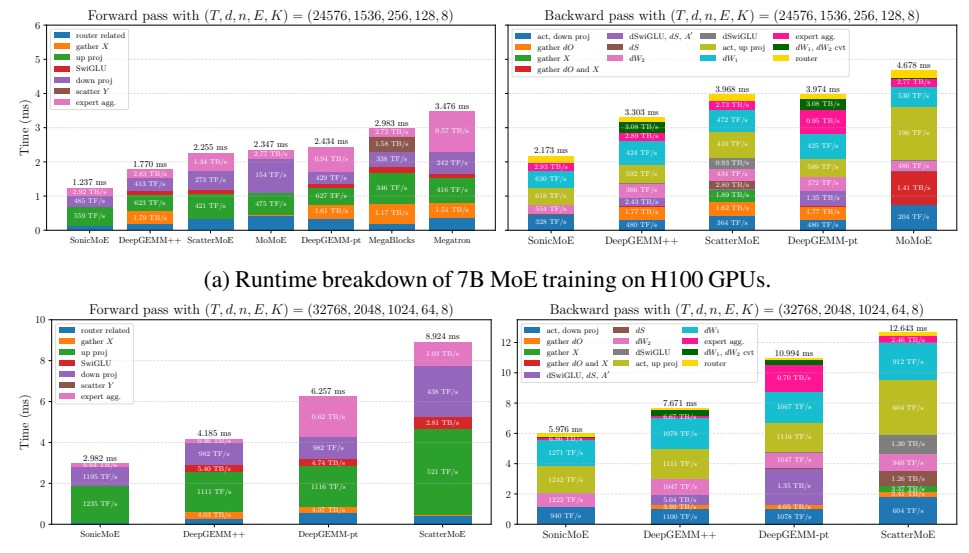

(a) Runtime breakdown of 7B MoE training on H100 GPUs.

(b) Runtime breakdown of OLMoE-sized 7B (Muennighoff et al., 2025) MoE training on B300 GPUs.

Figure 2: Runtime breakdown of different MoE kernels (ms ↓) on H100 and B300 GPUs. We annotate the model memory bandwidth (TB/s ↑) for memory-bound kernels and compute throughput (TFLOPS ↑, abbr as TF/s in the figure) for grouped GEMM kernels. We use the GroupedMLP for Megatron, and ParallelDroplessMLP for MegaBlocks. DeepGEMM does *not* provide an efficient router implementation, gather and expert aggregation kernels during the forward pass, where we use a standard PyTorch implementation ("DeepGEMM-pt") or our optimized kernels ("DeepGEMM++") for them.

The expressivity of fine-grained MoE comes from the diversity of every token's expert selection, which in turn leads to linearly-scaled IO cost w.r.t. expert granularity. To sustain high throughput, we need to maximally (1) reduce IO access via fusion and (2) overlap the IO latency with compute. We first examine the token gather fusion with computation, and math and IO fusion with epilogue in Section 4.1 and 4.2 respectively. We then describe the techniques to overlap MMA with IO in Section 4.3. In Appendix B, we compare SonicMoE with other MoE kernel designs with a summary in Table 4.

## 4.1 GATHER FUSION WITH HBM LOAD

SonicMoE builds on efficient varlen-$M$ and varlen-$K$ Grouped GEMM kernels with two key forms of fusion: gather fusion during input loading from global memory (GMEM, often the HBM) to shared memory (SMEM), and epilogue fusion for elementwise operations. Both reduce unnecessary memory traffic and create additional opportunities to overlap IO with MMA. In particular, gather fusion is critical because many baseline implementations still require a separate gather stage before GEMM, which introduces substantial IO overhead in fine-grained MoE training as illustrated in Figure 2a.

## 4.2 EPILOGUE FUSION

We exploit the epilogue to reduce unnecessary IO accesses with the following design choices:

- **SwiGLU and dSwiGLU fusion**: We fuse the SwiGLU and backward of SwiGLU with the epilogue of forward up-proj and backward down-proj activation gradient kernel respectively (Costin et al., 2025).
- **Computing $dH$ and $dS$ in backward down-proj activation gradient ($dH$) kernel's epilogue**: in Appendix D.1, we show that SonicMoE's $dS = \langle dA, A' \rangle$ is the computationally and activation memory-efficient choice for fine-grained MoEs. Our $dH$ kernel produces the same output with far less total time than ScatterMoE's down-proj act, $dS$, and dSwiGLU combined together in Figure 2a and 2b.

## 4.3 GEMM MMA OVERLAPPING WITH ASYNCHRONOUS IO

**Hopper GPUs.** On NVIDIA Hopper GPUs, GEMM is performed asynchronously with a producer-consumer paradigm (Shah et al., 2024). We can overlap the IO of 1 warpgroup with GEMM of another warpgroup with a smaller tile size. Once this is finished, we switch the roles of the warpgroups (effectively interleaving IO and GEMM). This is often referred to as *Ping-Pong scheduling* (Wright & Hoque, 2024b; Shah et al., 2024) on Hopper GPUs in Figure 9. Ping-Pong scheduling is particularly useful to maintain high Tensor Core throughput with heavy epilogue. For example, in the $dH$ kernel's epilogue, we need to load $H$ and execute multiple activation and reduction operations. Besides Ping-Pong scheduling, SonicMoE also relies on asynchronous TMA operations to perform asynchronous IO operations to overlap with the MMA operations:

- **Asynchronous TMA load during $dH$ kernel's epilogue**: In the $dH$ kernel's epilogue, we create a dedicated pipeline to load $H$ asynchronously, as illustrated in Figure 9.
- **Asynchronous TMA store in forward $Y$ and backward $d\tilde{X}$ kernel**: SonicMoE does *not* fuse the scatter with HBM store for both kernels. This is primarily because the scatter fusion requires a synchronous store instruction on Hopper GPUs.

**Blackwell GPUs.** On NVIDIA Blackwell GPUs, GEMM kernels use the same "Ping-Pong" scheduling in spirit, but the implementation differs from Hopper. Blackwell introduces Tensor Memory (TMEM), a dedicated on-chip memory per SM (NVIDIA, 2025b). The MMA accumulator results are stored directly in TMEM which enables a two-stage accumulator pipeline. While one stage performs MMA operations, the other stage executes the epilogue. The new Blackwell's MMA instruction, UMMA, is a single-threaded asynchronous operation that enables epilogue warps to read and process results from one TMEM stage concurrently with MMA warps accumulating into the other stage.

## 5 TOKEN ROUNDING ROUTING

In this section, we analyze the hardware efficiency under sparse MoE training regime and identify that as MoEs become sparser, the wasted compute on padded GEMM tiles accumulate to a nontrivial

amount, known as "tile quantization" effects. In response, we propose a novel routing method "token rounding" to eliminate tile quantization effects.

## 5.1 TRAINING EFFICIENCY OF SPARSE MoE

---

**Algorithm 4** Token rounding routing

---

**Input** : number of experts $E$ and expected activated number of experts $K$ per token; tile size $M_{\text{tile}}$; router scores $S \in [0,1]^{T \times E}$. round_and_sparsify that determines rounding up or down.

**Output** : $M_{\text{tile}}$-rounded routed token list and scores $\{\pi'_e, s'_e\}_{e \in [E]}$

**(1) Top-$K$ token choice sorting**
$(S_{\text{topK}}, I_{\text{topK}}) \leftarrow \text{TopK}(S, K)$

**(2) Calculate each expert's received token frequencies and its $M_{\text{tile}}$-rounded multiples**
$(f_e, \lceil f_e \rceil_{M_{\text{tile}}}, \lfloor f_e \rfloor_{M_{\text{tile}}})$
$\leftarrow (\sum_t \mathbf{1}_{\{e \in I_{\text{topK},t}\}}, \lceil f_e/M_{\text{tile}} \rceil \cdot M_{\text{tile}}, \lfloor f_e/M_{\text{tile}} \rfloor \cdot M_{\text{tile}})$

**(3) Build Top-$K$-preferred $S'$ for expert-wise ranking**
// ensure non-top-$K$ entries are smaller
$S'_e \leftarrow S_e - 1$
**for** $t \in [T]$ & $k \in [K]$ *in parallel* **do**
$\quad S'_{t, I_{\text{topK}}(t,k)} \leftarrow S_{\text{topK},t,k}$

**(4) Token rounding per expert**
**for** $e \in [E]$ **do**
$\quad$ // token ordering and sorted scores
$\quad \pi_e, s_e \leftarrow \text{sort}(S'_e)$
$\quad$ // token rounding for expert $e$
$\quad \pi'_e, s'_e \leftarrow \text{round\_and\_sparsify}(\pi_e, s_e, f_e, \lceil f_e \rceil_{M_{\text{tile}}}, \lfloor f_e \rfloor_{M_{\text{tile}}})$

---

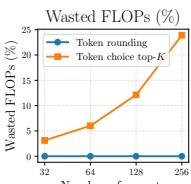

Figure 3: Wasted FLOPs by padding during MoE forward & backward pass with $T = 16k$, $d = 4k$, $n = 1024$, $K = 4$ as illustrated in the bottom right 2 subfigures of Figure 7.

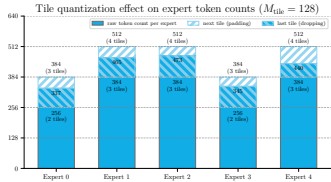

Figure 4: A demonstration of tile quantization effect for sparse MoE. The rounding subroutine in TR makes a binary decision for discarding or padding tokens to guarantee that each expert receives an $M_{\text{tile}}$-multiple number of tokens.

Besides granularity, the arithmetic intensity of MoE also depends on the MoE activation ratio $\rho$ as shown in Equation 4. When we scale down $\rho$, the expected number of received tokens per expert will also linearly decrease and the GEMM computation shifts towards a memory-bound regime.

**Tile quantization effect.** GEMM on modern GPUs is often computed in tiles (NVIDIA, 2022) and we always need to pad to the next tile-sized multiples if any dimensions of $\mathbf{M}, \mathbf{N}, \mathbf{K}$ are not fully divisible by tile sizes. Once the size of input (e.g. token dimension per expert) is small, the wasted TFLOPS by padding can be nontrivial. Therefore, we propose to use token rounding to avoid launching such extra tiles, thereby leading to more efficient training. We also empirically show that our token rounding method does not affect model quality while achieving much higher training throughput.

## 5.2 TOKEN ROUNDING ROUTING

As such, we introduce token rounding (TR), a two-step sorting procedure shown in Algorithm 4. TR first computes the vanilla token-choice (TC) routing result and then sorts each expert's assigned tokens by router score, similar to the EC sorting step. It then either discards some TC-selected tokens or pads additional tokens from the second sorting step. Between these two steps, we process the routing weight matrix so that TC-selected tokens are always preferred to EC-selected tokens, ensuring that any discarding or padding affects only the last input tile of each expert.

TR relies on a round_and_sparsify subroutine to decide between discarding and padding. By default, we round each expert frequency to the nearest multiple of $M_{\text{tile}}$: we pad EC-selected tokens when $\lceil f_e \rceil_{M_{\text{tile}}} - f_e < f_e - \lfloor f_e \rfloor_{M_{\text{tile}}}$. Table 6 shows that TR is robust to the choice of rounding subroutine, and that this simple nearest-rounding rule already yields strong task performance; additional discussion of alternative rounding rules is provided in Appendix E.2.

TR guarantees that, for each expert, the maximum deviation from TC routing is at most one tile. Despite this small perturbation, it remains robust in sparse MoE training, serves as an effective substitute for TC (Table 1), and is generally stable when $\bar{T}_e/M_{\text{tile}} \geq 2$ (Tables 7 and 8). At the same time, by eliminating tile-quantization waste, TR consistently improves training throughput over vanilla TC in highly sparse regimes, achieving up to 16% higher TFLOPS as $E$ increases while $K$ remains fixed (Section 6.3.2).

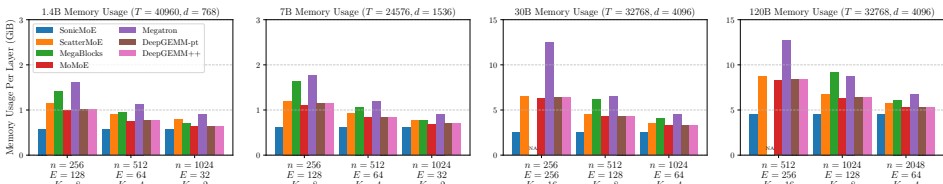

Figure 5: Peak activation memory usage per layer across different model scales (1.4B–120B). MegaBlocks does not support small $n$. The benchmark configurations are listed in Table 9. We only cache $X$, gathered $X_e$, $H_e$ for each expert $e$ and routing metadata which is the minimum required for backward computation without GEMM recomputation for DeepGEMM-pt and DeepGEMM++. Their definitions are the same as Figure 2a.

# 6 EXPERIMENTS

We now evaluate the three main components of SonicMoE. First, we compare activation memory usage against prior MoE implementations in Section 6.1. Next, we evaluate training throughput in Section 6.2. Finally, we show the token rounding routing preserves downstream quality in Section 6.3.1 and improves training efficiency in sparse MoE training configurations in Section 6.3.2.

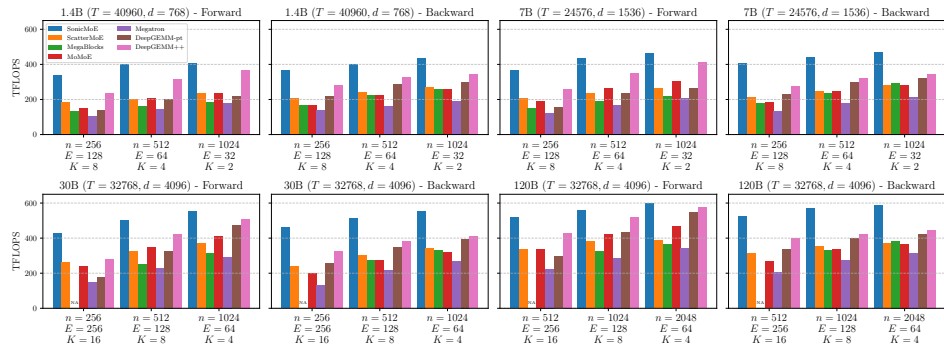

(a) Forward & backward TFLOPS for different MoE kernels on H100 GPUs.

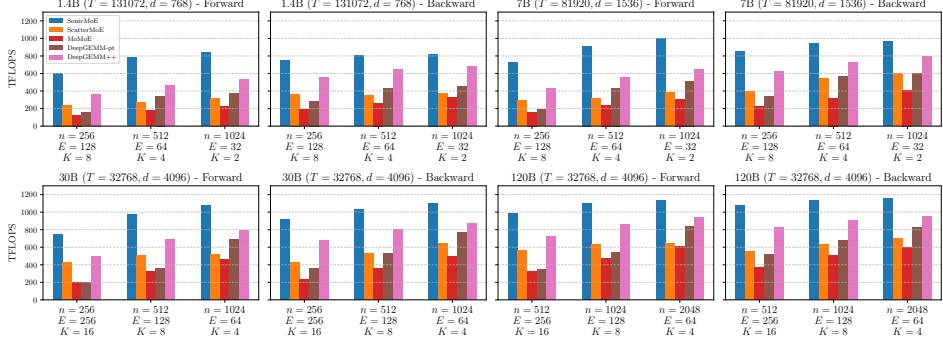

(b) Forward & backward TFLOPS for different MoE kernels on B300 GPUs.

Figure 6: Forward & backward TFLOPS for different MoE kernels on H100 and B300 GPUs.

## 6.1 SONICMOE'S ACTIVATION MEMORY

We demonstrate that the peak activation memory for SonicMoE has the lowest activation memory footprint for a single MoE layer as shown in Figure 5 across all scales. For the 7B model with $n = 256$, our approach reduces memory usage by **45%** compared to ScatterMoE, and more significantly compared to MoMoE. For 30B and 120B models, the gap becomes even wider: at 120B scale, our method saves more than **3GiB memory per layer** compared to MoMoE. We also validate that SonicMoE's activation memory stays constant w.r.t. expert granularity as shown in Figure 1.

## 6.2 SONICMOE'S TRAINING THROUGHPUT

Figure 6a reports the compute throughput of forward and backward passes of one MoE layer on H100 GPUs. Across all model scales, our method consistently achieves the highest TFLOPS. In 1.4B and 7B settings, our approach improves TFLOPS by 40% compared to ScatterMoE and MoMoE. This throughput gap becomes wider for 30B and 120B when other baselines either fail to support certain $n$ sizes (MegaBlocks) or suffer from significant performance degradation (MoMoE).

**We further measure the real training throughput of a 7B MoE model with FSDP-2 on H100 GPUs: SonicMoE on 64 H100s gets 213 billion tokens per day which achieves similar throughput as ScatterMoE on 96 H100s with 225 billion tokens per day.** The throughput for this is measured using the lm-engine codebase (Mishra, 2024). We shard the model using ZeRO-3 within a single node (8x H100s) and replicate this sharded unit across nodes for this experiment.

Besides H100 GPUs, we also measure SonicMoE's compute throughput on B300 GPUs in Figure 6b. We mainly compare with DeepGEMM++ which is powered by DeepGEMM SM100 BF16 grouped GEMM kernels. SonicMoE still demonstrates an overall speedup over DeepGEMM++, and such speedup is more pronounced when we increase expert granularity similar to the trend on H100 GPUs. For example, when we increase the expert granularity $d/n$ from 2 to 8 for 120B MoE training, SonicMoE achieves a greater relative speedup over DeepGEMM++ from 20.9% and 22.1% in forward and backward to 35.2% and 30.9% respectively.

## 6.3 TOKEN ROUNDING

### 6.3.1 TOKEN ROUNDING'S GENERAL TASK EVALUATION

Table 1: Comparison of different routing methods' task evaluation. "Train" and "Val" refer to the perplexity at the last 100 steps of training and at the validation set respectively. The next 11 columns are downstream tasks evaluated at the end of training and we report the accuracy for each. "Avg" is the mean accuracy across these 11 downstream tasks. We use TC top-$K$ routing for TR, EC, and token dropping baselines when evaluating validation perplexity and task performance. $\bar{T}_e$ means the average number of received tokens in each microbatch per expert.

(a) **0.5B params, 40B tokens, 2/64 activated** ($\bar{T}_e = 512$, $M_{\text{tile}} = 128$)

| | Train | Val | | | | | | | | | | | | Avg |
|---|---|---|---|---|---|---|---|---|---|---|---|---|---|---|
| TR | **16.22** | **15.92** | 51.4 | 41.6 | 78.4 | **65.4** | 31.6 | **38.1** | 65.0 | 31.0 | 61.1 | 57.4 | 29.1 | 50.0 |
| TC top-$K$ | 16.34 | 15.94 | 51.0 | **41.9** | 78.5 | 64.8 | 33.0 | **38.1** | **67.0** | 30.8 | 54.7 | 55.8 | 30.1 | 49.6 |
| TC (token drop) | 16.44 | 16.10 | 51.1 | 41.4 | 78.7 | 64.9 | 31.6 | 38.0 | 62.0 | **32.8** | **61.9** | **58.9** | **30.8** | 50.2 |
| EC | 16.83 | 18.61 | 49.6 | 41.4 | 79.1 | 64.4 | **33.4** | 36.9 | 62.0 | **32.8** | 60.2 | 55.8 | 29.1 | 49.5 |
| EC (aux router) | 16.80 | 21.80 | 50.0 | 40.9 | 75.2 | 63.7 | 28.2 | 35.2 | 61.0 | 31.5 | 57.2 | 53.3 | 24.7 | 47.4 |
| EC (ft TC router) | 16.81 | 16.98 | 50.0 | 41.7 | **79.7** | 64.9 | 31.6 | 36.8 | 63.0 | 32.1 | 60.7 | 54.6 | 27.4 | 49.3 |

(b) **1.8B params, 40B tokens, 8/256 activated** ($\bar{T}_e = 512$, $M_{\text{tile}} = 128$)

| | Train | Val | | | | | | | | | | | | Avg |
|---|---|---|---|---|---|---|---|---|---|---|---|---|---|---|
| TR | **13.34** | **13.10** | 53.4 | 42.1 | 81.7 | 69.6 | **35.2** | **45.3** | 70.0 | 33.2 | **61.4** | 63.0 | **33.4** | **53.5** |
| TC top-$K$ | 13.51 | 13.12 | 50.1 | **42.9** | 81.3 | **69.8** | 33.8 | 45.2 | **71.0** | **34.1** | 56.7 | **64.6** | 31.1 | 52.8 |
| TC (token drop) | 13.62 | 13.19 | **55.4** | 41.6 | **82.2** | 68.6 | 34.0 | 45.0 | 69.0 | 34.0 | 54.4 | 63.5 | 31.4 | 52.7 |
| EC | 14.92 | 19.82 | 51.9 | 40.8 | 77.7 | 65.8 | 30.0 | 39.8 | 67.0 | 30.9 | 60.7 | 56.0 | 28.4 | 49.9 |
| EC (aux router) | 14.94 | 18.01 | 50.6 | 41.8 | 79.8 | 65.8 | 31.6 | 39.3 | 62.0 | 31.8 | 59.7 | 55.8 | 29.8 | 49.8 |
| EC (ft TC router) | 14.81 | 15.01 | 52.7 | 41.1 | 79.6 | 66.9 | 30.6 | 40.2 | 66.0 | 31.9 | 60.5 | 57.2 | 30.8 | 50.7 |

(c) **1.4B params, 100B tokens, 2/128 activated** ($\bar{T}_e = 512$, $M_{\text{tile}} = 128$)

| | Train | Val | | | | | | | | | | | | Avg |
|---|---|---|---|---|---|---|---|---|---|---|---|---|---|---|
| TR | **13.31** | **13.22** | 52.8 | 41.8 | 80.8 | **68.7** | 33.0 | **43.4** | 67.0 | 33.6 | **60.2** | 60.7 | 29.8 | **52.0** |
| TC top-$K$ | 13.50 | 13.32 | 51.3 | 42.0 | **83.2** | 68.2 | **34.0** | **43.4** | 66.0 | **35.4** | 57.9 | **61.6** | 29.4 | **52.0** |
| TC (token drop) | 13.35 | 13.29 | 50.0 | **42.2** | 81.7 | 68.3 | 31.2 | 43.3 | 66.0 | 34.3 | 56.6 | 59.5 | **30.8** | 51.3 |
| EC | 14.08 | 24.79 | 51.5 | 41.7 | 81.0 | 66.1 | 33.2 | 40.6 | 64.0 | 34.0 | 56.3 | 56.5 | 27.4 | 50.2 |
| EC (aux router) | 14.01 | 37.52 | 49.7 | 40.2 | 73.6 | 57.5 | 27.6 | 33.2 | 61.0 | 27.8 | 58.8 | 45.2 | 24.2 | 45.3 |
| EC (ft TC router) | 14.24 | 14.75 | 52.2 | 42.6 | 79.4 | 65.7 | 32.8 | 40.8 | 64.0 | 34.9 | 58.3 | 57.2 | 27.1 | 50.5 |

In this section, we assess the quality of trained MoEs using our token rounding algorithm. We use TR for training and during evaluation we switch to token-choice top-$K$ routing. This assesses whether TR can be replaced by TC after training. We use the OLMoE codebase and construct MoE models with OLMoE base architecture (Muennighoff et al., 2025). We use a deduplicated version of FineWeb-Edu (Ben Allal et al., 2024) for training all our models. More details are included in Appendix G.

In Table 1, we consistently use softmax renormalization for TR. We compare TR to token-choice (TC) top-$K$ routing and expert-choice (EC) routing (Zhou et al., 2022). We also consider MoD's

approach (Raposo et al., 2024) that trains an auxiliary router to predict the EC router's selection during inference. This baseline is referred to as "EC (aux router)" in each subtable in Table 1. We also adapt EC routing to TC routing by finetuning a learned TC top-$K$ router and compare its task performance against TR's task performance *without any adaptation*. This is the "EC (ft TC router)" baseline in Table 1. Finally, we consider a token dropping baseline in which we set the capacity of each expert as the largest multiple of $M_{\text{tile}}$ not exceeding its token frequencies and we discard the tokens with lowest scores. This is the "TC (token drop)" baseline.

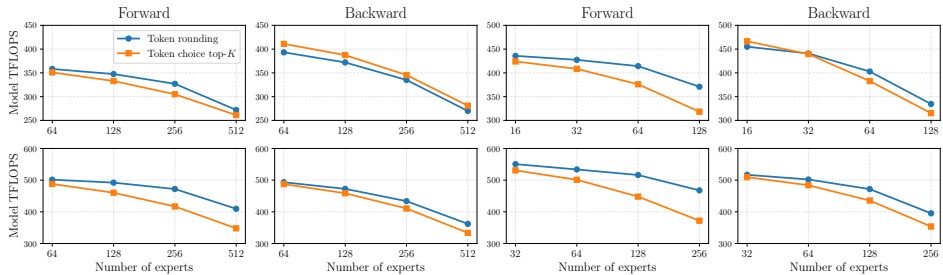

Figure 7: Forward & backward model TFLOPS for SonicMoE MoE kernels with different routing methods on H100 GPUs. We compare TR equipped with "nearest rounding to $M_{\text{tile}}$-multiples via expert frequency" subroutine against TC top-$K$ routing. Configuration details are in Appendix F.

**TR's train-test gap.** Across all sparse MoE configurations in Table 1, we consistently observe similar model quality between TR and TC. In fact, TR achieves slightly lower validation perplexity and higher average accuracy under the extremely sparse MoE settings for the 1b and 1c. There is a noticeable discrepancy between EC and TC as the train and val PPL for EC can have $\gtrsim 3$ gap for 1b and 1c compared to TC and TR's usual $\lesssim 0.3$ gap. TC finetuning is more effective than the auxiliary router to close this gap, but TR's task evaluation is still always better. In this case, TR can serve as an in-place substitute for TC during training.

### 6.3.2 TOKEN ROUNDING'S TRAINING THROUGHPUT

In Figure 7, we benchmark the token rounding's MoE main kernel runtime (without router) against top-$K$ token choice routing. We focus on the iso-FLOPs setting by keeping $T$, $n$ and $K$ constant. We linearly increase the number of experts $E$ while keeping $K$ constant to increase MoE sparsity. As we linearly increase $E$, we observe a drop in TFLOPS for token-choice routing. This is due to the (1) tile quantization effect as the wasted FLOPs spent on padding roughly linearly increases with the MoE sparsity as shown in Figure 3 and (2) the linearly increased IO due to more expert weights. We observe a drop in both TC and TR TFLOPS as we increase $E$, but the drop is more pronounced for TC.

For the $3^{\text{rd}}$ and $4^{\text{th}}$ columns in the top row in Figure 7, an MoE model with 128 experts ($K/E = 1/64$) and $n = 1k$ with token rounding routing achieves 16.5% model TFLOPS improvement in forward and 6.1% in backward resulting in an end-to-end improvement of 9.4%. In general, **we observe that as we move to larger intermediate sizes (more compute-bound) and higher MoE sparsity, the gap between TR and TC top-$K$ becomes larger.**

## 7 CONCLUSION

We present SonicMoE, a co-design solution that jointly optimizes MoE architecture and GPU kernels to address the training challenges of granular and sparse MoEs. Our contributions include: (1) a memory-efficient algorithm that minimizes activation size as MoEs become more fine-grained, (2) GPU kernels that overlap IO with computation for throughput improvement, and (3) tile-aware token rounding that yields additional speedup without quality loss. Future directions include extending to low-precision and microscaling formats (FP8, MXFP8, MXFP4) for further memory savings, and overlapping communication with computation in distributed settings like expert parallelism. We envision future model architecture designs that optimize for quality per compute hour rather than just quality per FLOP—jointly considering algorithmic and hardware efficiency.

ACKNOWLEDGMENT

We gratefully acknowledge the support of Schmidt Sciences AI2050 fellowship, the Google ML and Systems Junior Faculty Awards, and the Google Research Scholar program. We also thank the Princeton Language Intelligence program for the computing resources support. We thank Shawn Tan for his generous support on our experiments. We also thank Songlin Yang, Yilong Zhao, Bharat Runwal, Xinyu Yang, Andrew Sheinberg, Lijie Yang, Yongye Zhu, Zhuoqing Song and numerous anonymous reviewers for providing valuable feedback.

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

## APPENDIX

In Section A, we summarize the notation used throughout the paper. In Section B, we compare SonicMoE with existing open-source MoE kernel designs. In Section C, we collect the referenced tables, figures, and the up-projection backward algorithm cited in the main paper. In Section D, we derive the gradient computation used by SonicMoE, and in Section D.1, we explain why our choice of computing $dS$ is both activation-memory-efficient and computationally efficient. In Section E, we present additional experiments, including the effect of expert granularity and ablations of token rounding with respect to the rounding subroutine, microbatch size, and tile size. In Section F, we provide the benchmark configurations used for the activation-memory and throughput results. Finally, in Section G, we include the model-training and evaluation details.

# A NOTATIONS

In Table 2, we describe the notations used in this paper.

Table 2: Notations and their explanations

| Notations | Explanation |
|---|---|
| $T$ | number of tokens in a microbatch |
| $d$ | model embedding dimension (hidden size) |
| $n$ | each expert's intermediate dimension |
| $E$ | total number of experts |
| $K$ | number of activated experts |
| $\rho$ | $\rho = K/E$ represents MoE activation ratio |
| $\bar{T}_e$ | $\bar{T}_e = \mathbb{E}_{e \in [E]}[T_e] = T\rho$ represents the expected number of received tokens in each microbatch by expert $e$ |
| $G$ | $G = d/n$ represents the MoE expert granularity. Greater $G$ means a MoE is more fine-grained |
| $\mathbf{M, N, K}$ | Dimensions for GEMM in CUTLASS. We define $A \in \mathbb{R}^{\mathbf{M} \times \mathbf{K}}$, $B \in \mathbb{R}^{\mathbf{K} \times \mathbf{N}}$, and $C \in \mathbb{R}^{\mathbf{M} \times \mathbf{N}}$ for $AB = C$ |
| $M_{\text{tile}}, N_{\text{tile}}, K_{\text{tile}}$ | tile size of $\mathbf{M}, \mathbf{N}, \mathbf{K}$ dimension for a single GEMM tile |
| $R_e$ | tile quantization residue $R_e := T_e \bmod M_{\text{tile}}$ |
| $X$ | $X \in \mathbb{R}^{T \times d}$, input token embeddings for an MoE layer |
| $W_1$ | $W_1 \in \mathbb{R}^{E \times d \times 2n}$, weight of up projection |
| $W_2$ | $W_2 \in \mathbb{R}^{E \times n \times d}$, weight of down projection |
| $\pi$ | $\pi \in \{0,1\}^{T \times E}$, a binary-valued matrix where $\pi_{t,e}$ represents if token $t$ is routed to expert $e$ |
| $S$ | $S \in \mathbb{R}^{T \times E}$, router scores. In practice, we only materialize the sparsified $S$ instead of the full $S$ |
| $H$ | $H \in \mathbb{R}^{TK \times 2n}$, output of up projection |
| $A$ | $A \in \mathbb{R}^{TK \times n}$, output of SwiGLU |
| $Y$ | $Y \in \mathbb{R}^{TK \times d}$, output of down projection |
| $O$ | $O \in \mathbb{R}^{T \times d}$, output of expert aggregation, also output of the entire MoE layer |
| $dO$ | $dO \in \mathbb{R}^{T \times d}$, activation gradient for $O$ |
| $dA'$ | $dA' = dO\, W_2^\top \in \mathbb{R}^{T \times n}$, GEMM output of $dO$ and $W_2$. Intermediate result for computing $dA$ and $dS$ |
| $dA$ | $dA = \text{Broadcast}(\mathbf{s})\, dA' \in \mathbb{R}^{T \times n}$, activation gradient for $A$ |
| $dY$ | $dY = \text{Broadcast}(\mathbf{s})\, dO \in \mathbb{R}^{TK \times d}$, activation gradient for $Y$. $dY$ is not used in SonicMoE. |
| $dS$ | $dS \in \mathbb{R}^{T \times E}$, activation gradient for $S$ |
| $A'$ | $A' = \text{Broadcast}(\mathbf{s})\, A \in \mathbb{R}^{T \times n}$, intermediate result and input for computing $dW_2$ |
| $dH$ | $dH \in \mathbb{R}^{T \times 2n}$, activation gradient for $H$ |
| $d\tilde{X}$ | $d\tilde{X} \in \mathbb{R}^{TK \times d}$, activation gradient for $X$ before aggregation |
| $dX$ | $dX \in \mathbb{R}^{T \times d}$, activation gradient for $X$ after aggregation |
| $dW_1$ | $dW_1 \in \mathbb{R}^{E \times d \times 2n}$, weight gradient for $W_1$ |
| $dW_2$ | $dW_2 \in \mathbb{R}^{E \times n \times d}$, weight gradient for $W_2$ |
| $A$ kernel | forward up-proj kernel |
| $Y$ kernel | forward down-proj kernel |
| $O$ kernel | forward expert aggregation kernel where each token aggregates all routed expert's result as the final forward output |
| $dH$ kernel | backward down-proj activation gradient kernel |
| $dW_2$ kernel | backward down-proj weight gradient kernel |
| $d\tilde{X}$ kernel | backward up-proj activation gradient kernel |
| $dW_1$ kernel | backward up-proj weight gradient kernel |
| $dX$ kernel | backward expert aggregation kernel where each token aggregates the routed expert's $d\tilde{X}$ |
| $\lceil f_e \rceil_{M_{\text{tile}}}, \lfloor f_e \rfloor_{M_{\text{tile}}}$ | $M_{\text{tile}}$-rounded multiples of expert frequency $f_e$. $\lceil f_e \rceil_{M_{\text{tile}}} = \lceil f_e / M_{\text{tile}} \rceil \cdot M_{\text{tile}}$  $\lfloor f_e \rfloor_{M_{\text{tile}}} = \lfloor f_e / M_{\text{tile}} \rfloor \cdot M_{\text{tile}}$ |
| $\lfloor S \rceil_{M_{\text{tile}}}, \lfloor f_e \rceil_{M_{\text{tile}}}$ | $\lfloor f_e \rceil_{M_{\text{tile}}}$ is $M_{\text{tile}}$-rounded multiples of expert frequency $f_e$. $\lfloor f_e \rceil_{M_{\text{tile}}} \in \{\lceil f_e \rceil_{M_{\text{tile}}}, \lfloor f_e \rfloor_{M_{\text{tile}}}\}$, and $\lfloor S \rceil_{M_{\text{tile}}}$ is the score after rounding in Algorithm 4. |

## B    SONICMOE'S COMPARISON WITH EXISTING MOE KERNEL DESIGN

Existing efficient MoE kernels also frame MoE computation as a Grouped GEMM, but their ingredients are different from SonicMoE. Here we provide an overview (but not a complete list) of key differences:

- **ScatterMoE** (Tan et al., 2024)[4] implements gather fusion for varlen-**M** Grouped GEMM but not for varlen-**K** Grouped GEMM. ScatterMoE also does not overlap MMA computation with memory IO. Moreover, ScatterMoE is also built on older versions of Triton where TMA is not supported. ScatterMoE computes $dS$ as $dS = \langle dO, Y \rangle$ which requires caching $Y$. This results in large IO cost and activation memory requirement. ScatterMoE's both forward and backward pass have limited fusion and hence it is much slower than SonicMoE, especially for backward computation.

- **MoMoE** (Costin et al., 2025)[5] also implements the gather fusion for varlen-M but not varlen-**K** Grouped GEMM similar to ScatterMoE. Although fused with up-proj activation gradient, the $dS$ computation still utilizes $dS = \langle dO, Y \rangle$. Similar to ScatterMoE, MoMoE does not use TMA for IO.

- **MegaBlocks** (Gale et al., 2023) has multiple MoE implementations and we focus on ParallelDroplessMLP[6] which is built on top of block-sparse matrix multiplication[7]. ParallelDroplessMLP first gathers and pads the tokens and then launches block-sparse GEMM for up and down-proj. Then, it launches a scatter kernel before reducing across the expert results. These sparse matrix multiplications usually take a longer time than the highly-optimized Grouped GEMM, as shown in Figure 2a, and the gather and scatter kernel have a total IO cost of $8TKd$ bytes which can be a bottleneck for fine-grained MoEs. We consider MegaBlocks's ParallelDroplessMLP as a block-sparse GEMM baseline in our benchmark and find that MoE implemented via Grouped GEMMs often have a higher training throughput than MoEs implemented via block-sparse GEMM.

- **Megatron-LM** (Shoeybi et al., 2019) also has multiple MoE implementations and we focus on GroupedMLP[8], which uses Grouped GEMM[9] from the CUTLASS library (Corporation, 2025) with JIT epilogue fusion as the GEMM backend. Similar to DeepGEMM, GroupedMLP does not fuse gather with the HBM load. A recent memory-efficient patch[10] fuses $S$ weighting with SwiGLU computation during forward, and during backward which allows the PyTorch autograd engine (Paszke et al., 2019) to follow a similar computational path as SonicMoE.
  Megatron-LM also implements TEGroupedMLP[11] which launches 4 CUDA streams to execute a list of GEMM (without contiguously-packed inputs, and without a persistent tile scheduler). In this case, each expert independently launches a new GEMM kernel leading to "bubbles" on the CUDA streams. This leads to underutilization of the GPU resources. We empirically find that TEGroupedMLP runs slower than GroupedMLP and so we use GroupedMLP across all benchmarks.

- **DeepGEMM** (Zhao et al., 2025b) designs a Grouped GEMM kernel for contiguously-packed inputs. They also don't implement any other fusion for both SM90 (Hopper) and SM100 (Blackwell) BF16 Grouped GEMM. DeepGEMM specializes more on distributed training with expert parallelism (Zhao et al., 2025c), and it is common to launch a separate all2all kernel (Lepikhin et al., 2024) which is then followed by a contiguous Grouped GEMM. Both DeepGEMM SM90 and SM100 BF16 kernels assume that each expert receives a multiple of $M_{\text{tile}}$ tokens as it does not implement the TMA tensor descriptor online update during the Grouped GEMM computation. DeepGEMM's BF16 GEMM on SM90 also does not employ Ping-Pong scheduling.

Additionally, ScatterMoE and MoMoE are both implemented in Triton (Tillet et al., 2019) for the ease of development at the expense of losing full programmability of the asynchronous compute and memory IO of Hopper GPUs (NVIDIA, 2022). For example, they cannot implement fine-grained

---

[4] https://github.com/shawntan/scattermoe/blob/47b5e1502e5a10e82c8e5945d761b877849871e7/scattermoe/mlp.py#L51

[5] https://github.com/tilde-research/MoMoE-impl/blob/d6e2d683185bfe4030265c3ca062564356faa61e/momoe/momoe.py#L914

[6] https://github.com/databricks/megablocks/blob/78eea65fda01e638af36ae38853bc51efb04a4b4/megablocks/layers/dmoe.py#L18

[7] https://github.com/databricks/megablocks/blob/78eea65fda01e638af36ae38853bc51efb04a4b4/megablocks/layers/mlp.py#L308

[8] https://github.com/NVIDIA/Megatron-LM/blob/610a75ef3a4a80c2ce2da436c19244e5362978d4/megatron/core/transformer/moe/experts.py#L100

[9] https://github.com/fanshiqing/grouped_gemm/blob/main/csrc/grouped_gemm.cu

[10] https://github.com/NVIDIA/Megatron-LM/commit/2659630721ac87237c8cb772b1c2f1b34176f443

[11] https://github.com/NVIDIA/Megatron-LM/blob/610a75ef3a4a80c2ce2da436c19244e5362978d4/megatron/core/transformer/moe/experts.py#L746

control of asynchronous load and store during the GEMM's epilogue. They also cannot overlap MMA with heavy epilogue operations using Ping-Pong scheduling. It becomes increasingly important to overlap IO operations in epilogue when the GEMM computations are small in size (as in the case of fine-grained MoEs) to achieve high GPU utilization.

## C  REFERENCED TABLES, FIGURES, AND ALGORITHMS

In Table 3, we provide a trending overview for open-source frontier MoE models. SonicMoE's computational workflow is illustrated in Figure 8, and a high-level comparison between SonicMoE and related efficient MoE kernel designs is shown in Table 4. Figure 9 provides an example of SonicMoE employing Ping-Pong warpgroup scheduling on Hopper GPUs. We present SonicMoE's backward pass of up-proj algorithm in Algorithm 5.

Table 3: **MoE Scaling Trends:** Here, we show the activation ratio as experts activated per token $K$ / total experts $E$ and expert granularity is shown as model embedding dimension ($d$) / expert intermediate size ($n$) for frontier open source models. We do not include the shared experts for the MoE sparsity calculation. The trend indicates new open-source MoE models tend to be more granular and sparser.

| Model | Release date | Parameters | Expert activation ratio ($K/E$) | Expert granularity ($d/n$) |
|---|---|---|---|---|
| Mixtral 8x22B (Mistral, 2024) | 11/23 | 131B | 25.0% (2/8) | $6144/16384 = 0.38$ |
| DBRX (The Mosaic Research Team, 2024) | 03/24 | 132B | 25.0% (4/16) | $6144/10752 = 0.57$ |
| Phi-3.5-MoE (Microsoft, 2024) | 09/24 | 42B | 12.5% (2/16) | $4096/6400 = 0.64$ |
| OLMoE (Muennighoff et al., 2025) | 09/24 | 7B | 12.5% (8/64) | $2048/1024 = 2.00$ |
| Granite 3.1-MoE (Granite, 2024) | 12/24 | 3B | 20.0% (8/40) | $1536/512 = 3.00$ |
| **DeepSeek-V3** (DeepSeek-AI et al., 2024) | **12/24** | **671B** | **3.13% (8/256)** | $7168/2048 = 3.50$ |
| **Qwen3 MoE** (QwenLM, 2025) | **04/25** | **235B** | **6.25% (8/128)** | $4096/1536 = 2.67$ |
| **QWen3-30B-A3B** (Qwen, 2025) | **05/25** | **30.5B** | **6.25% (8/128)** | $2048/768 = 2.67$ |
| **Kimi K2** (Kimi et al., 2025) | **07/25** | **1.04T** | **2.08% (8/384)** | $7168/2048 = 3.50$ |
| gpt-oss-120b (OpenAI, 2025) | 08/25 | 120B | 3.13% (4/128) | $2880/2880 = 1.00$ |
| **GLM-4.5-Air** (Zeng et al., 2025) | **08/25** | **106B** | **6.25% (8/128)** | $4096/1408 = 2.91$ |
| **Qwen3-Next-80B-A3B-Instruct** (Qwen, 2025) | **09/25** | **81B** | **1.95% (10/512)** | $2048/512 = 4.00$ |
| **DeepSeek-V3.2-Exp** (DeepSeek-AI, 2025) | **10/25** | **685B** | **3.13% (8/256)** | $7168/2048 = 3.50$ |

| Features \ Methods | SonicMoE | ScatterMoE | MoMoE | MegaBlocks | Megatron | DeepGEMM |
|---|---|---|---|---|---|---|
| Gather fused with GMEM-to-SMEM (HBM) load (Sec. 4.1) | ✓ | fwd ✓, bwd ✗ | fwd ✓, bwd ✗ | ✗ | ✗ | ✗ |
| SwiGLU and dSwiGLU fused with epilogue (Sec. 4.2) | ✓ | ✗ | ✓ | ✗ | ✓ | NA |
| $dS$ computed as $\langle dA'_{e,t}, A_{e,t} \rangle$ (Sec. 4.2, App. D.1) | ✓ | ✗ | ✗ | ✗ | ✓ | NA |
| Epilogue that computes $dH$, $dS$ together (Sec. 4.2) | ✓ | ✗ | ✗ | ✗ | ✗ | NA |
| Overlap MMA with epilogue/IO (Sec. 4.3) | ✓ | ✗ | ✗ | ✗ | ✗ | ✗ |
| Do *not* need a separate scatter kernel | ✓ | ✓ | ✓ | ✗ | NA | NA |
| Efficient top-$K$ sorting | ✓ | ✗ | ✗ | ✗ | ✗ | NA |
| Do *not* need shape-alignment efforts outside GEMM kernels | ✓ | ✓ | ✓ | ✗ | ✗ | ✗ |

Table 4: Comparison between SonicMoE and prior MoE kernels. ✓ means that the kernel implements the feature or a functionality similar in semantics, and ✗ means the feature is missing from the kernel. "NA" means that the feature is out of the expected scope. We use the GroupedMLP for Megatron and ParallelDroplessMLP for MegaBlocks. More discussion is included in Appendix B.

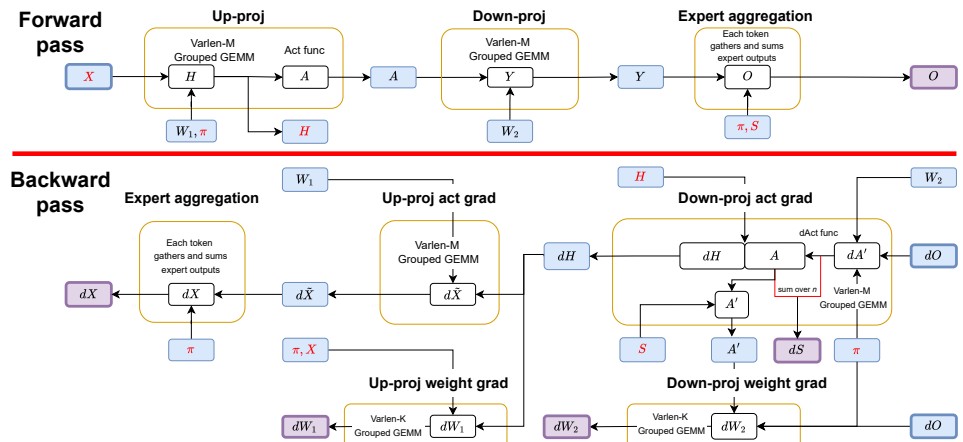

Figure 8: Computational workflow of SonicMoE's 8 launched kernels, grouped by yellow boxes. 3 and 5 kernels are launched during forward and backward computation respectively. The incoming arrows to a yellow circle indicate a variable loaded from HBM to SRAM, and an outgoing arrow represents a variable stored to HBM. We color the boxes of all variables on HBM, with purple boxes indicating the output of forward and backward while blue boxes indicate intermediate variables or weights ($W_1$, $W_2$). We color all cached activations $X$, $H$, $\pi$, $S$ in red. Algorithm 2 formally describes SonicMoE's forward pass, and Algorithm 3 and 5 describe the backward pass.

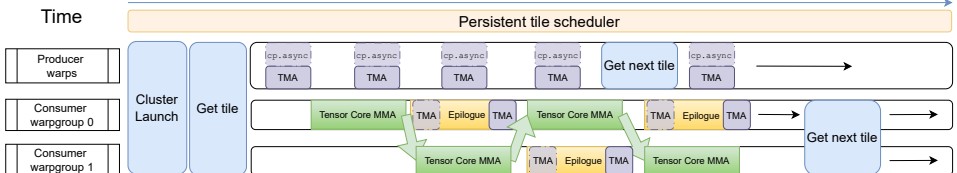

Figure 9: SonicMoE's Ping-Pong warpgroup scheduling on Hopper GPUs. The green arrows indicate that a consumer warpgroup signals the start of the epilogue and the other consumer warpgroup can proceed with the MMA. Once this step is complete, the roles of 2 consumer warpgroups is switched. SonicMoE mainly uses Ping-Pong for forward down-proj $Y$ kernel and backward down-proj activation gradient $dH$ kernel as they both have heavy epilogue. In $dH$ kernel, SonicMoE has an asynchronous TMA load during epilogue, and producer warps need to issue cp.async for gathering $dO$ and load expert weights with TMA. This figure is adapted from Wright & Hoque (2024a)'s blog on Ping-Pong scheduling.

---

**Algorithm 5** SonicMoE's MoE kernel backward pass of up-proj.

**Input** : $X, \pi, W_1, dH$

**Output** : $dX, dW_1$.

Up-proj act $d\tilde{X}$ kernel $(dH, W_1) \rightarrow d\tilde{X}$:

 **Parallel for** $e \in [E]$ **do**

  $dH_e, W_{1,e} \leftarrow \text{load}(dH_e, W_{1,e})$

  $d\tilde{X}_e \leftarrow dH_e W_{1,e}^\top$

  $d\tilde{X}_e \leftarrow \text{store}(d\tilde{X}_e)$

Up-proj weight $dW_1$ kernel $(X, dH, \pi) \rightarrow dW_1$:

 **Parallel for** $e \in [E]$ **do**

  $X, \pi_{:,e}, dH_e \leftarrow \text{load}(X, \pi_{:,e}, dH_e)$

  $X_e \leftarrow \text{Gather}(X, \pi_{:,e})$

  $dW_{1,e} \leftarrow X_e^\top dH_e$

  $dW_{1,e} \leftarrow \text{store}(dW_{1,e})$

Expert aggregation $dX$ kernel $(d\tilde{X}, \pi) \rightarrow dX$:

 **Parallel for** $t \in [T]$ **do**

  $d\tilde{X}, \pi_{t,e} \leftarrow \text{load}(d\tilde{X}, \pi_{t,e})$

  $dX_t \leftarrow \sum_{e \in [E]} \pi_{t,e} d\tilde{X}_{e,t}$

  $dX_t \leftarrow \text{store}(dX_t)$

## D GRADIENT COMPUTATION

For an expert $e$, let

$$X_e \in \mathbb{R}^{T_e \times d}, \quad W_{1,e} \in \mathbb{R}^{d \times 2n}, \quad W_{2,e} \in \mathbb{R}^{n \times d} \tag{5}$$

The forward activation computation is given by:

$$H_e = X_e W_{1,e} \in \mathbb{R}^{T_e \times 2n}, \qquad A_e = \text{SwiGLU}(H_e) \in \mathbb{R}^{T_e \times n}, \qquad Y_e = A_e W_{2,e} \in \mathbb{R}^{T_e \times d}. \tag{6}$$

The token aggregation with scores $S = \{s_{t,e}\}$ is given by

$$O_t = \sum_e s_{t,e} Y_{e,t}, \qquad dO_t \in \mathbb{R}^{1 \times d} \text{ as the gathered results from } dO. \tag{7}$$

We know

$$dY_{e,t} = s_{t,e} \, dO_t \quad \implies \quad dY_e = \text{Broadcast}(\mathbf{s}_e) \, dO_e. \tag{8}$$

Define the Grouped GEMM output as $dA'_e := dO_e \, W_{2,e}^\top \in \mathbb{R}^{T_e \times n}$.

Then from Equation 8

$$dA_e = dY_e \, W_{2,e}^\top = \text{Broadcast}(\mathbf{s}_e) \, dA'_e. \tag{9}$$

The activation gradient for score $dS$ is

$$\boxed{dS_{t,e} = \langle dO_t, Y_{e,t} \rangle = \langle dO_t \, W_{2,e}^\top, A_{e,t} \rangle = \langle dA'_{e,t}, A_{e,t} \rangle}. \tag{10}$$

In addition, we can derive $dH_e$ from $dA_e$ and $A_e$ (recomputed from $H_e$) as:

$$dH_e = \text{dSwiGLU}(dA_e, H_e). \tag{11}$$

Using Equation 8,

$$dW_{2,e} = A_e^\top \, dY_e = A_e^\top \left( \text{Broadcast}(\mathbf{s}_e) \, dO_e \right) = \underbrace{\left( \text{Broadcast}(\mathbf{s}_e) \, A_e \right)}_{A'_e}^\top dO_e. \tag{12}$$

### D.1 COMPUTATIONAL CHOICES FOR $dS$

If we do not implement custom kernels and rely solely on PyTorch's autograd (AD) engine, we can add the expert weighting ($S$) either (1) before down-proj forward or (2) after down-proj forward. Both yield identical results for forward and backward, but the computation for $dS$ is different. For (1), we need to compute $\langle dA'_{e,t}, A_{e,t} \rangle$ which is used by SonicMoE and Megatron, ScatterMoE, and MegaBlocks compute $\langle dO_t, Y_{e,t} \rangle$ as required in (2).

Note that $dS$ can be computed as any of $dS_{t,e} = \langle dA'_{e,t}, A_{e,t} \rangle = \langle dO_t, Y_{e,t} \rangle$, however computing it as $dS_{t,e} = \langle dA'_{e,t}, A_{e,t} \rangle$ is a computationally and activation memory-efficient choice due to the following reasons:

- **Additional HBM traffic (0 vs. $2TKd$ bytes)**: $\langle dA'_{e,t}, A_{e,t} \rangle$ requires $dA'_{e,t}$ and $A_{e,t}$ are already computed during the $dH$ kernel, so we can avoid extra unnecessary loads.
- **Extra cached activation memory (0 vs. $2TKd$ bytes)**: One of the reasons why the cached activation memory for ScatterMoE, MoMoE and MegaBlocks fails to stay constant w.r.t. expert granularity is the required caching of $Y$ for computing $dS$.
- **Parallel reduction rounds ($\log_2(n)$ vs. $\log_2(d)$)**: $\langle dA'_{e,t}, A_{e,t} \rangle$ reduces over $n$ while $\langle dO_t, Y_{e,t} \rangle$ reduces over $d$. This difference saves at least $\log_2(d/n)$ rounds of reduction.

# E    MORE EXPERIMENTS

In this section, we investigate the qualitative improvement from fine-grained MoE in Section E.1. We also investigate the effect of rounding subroutines round_and_sparsify in Algorithm 4 and the effects from microbatch size $T$ and tile size $M_{\text{tile}}$ for token rounding in Section E.3.

## E.1    EFFECT OF EXPERT GRANULARITY

Here we validate the effectiveness of adopting fine-grained MoE. We fix the MoE activation ratio $\rho = K/E$ for the 0.5B and 1.4B model and we proportionally scale up $K$ and $E$ while linearly decreasing $n$ from row 1 to row 3 in Table 5a and 5b.

In general, we observe a better performance for $n = 256$ than $n = 1024$ which is also consistent with the MoE scaling trends mentioned in Table 3. In Figure 1 right subfigure, we find both SonicMoE and cuBLAS can still sustain the throughput from $n = 1024$ to $n = 256$ under iso-FLOPs, but starting from $n = 256$ FLOPs will drop linearly w.r.t. granularity. Therefore, we choose $n = 256$ for all experiments in Table 1.

Table 5: Evaluation of MoE w.r.t. granularity with iso-FLOPs ($nK$ is constant) and iso-params ($nE$ is constant) settings. "PPL" refers to the validation perplexity at the end of training. "Avg" is the mean accuracy across the 11 downstream tasks. The "dense, iso-FLOPs" refers to a dense model with $nK$ as the intermediate size, while the "dense, iso-params" refers to a dense model with $nE$ as the intermediate size.

(a) **0.5B params, 20B tokens, 8/64 activated**

| $(E,K,n)$ | PPL | Wino | SIQA | SciQ | PIQA | OBQA | HS | COPA | CSQA | BoolQ | ArcE | ArcC | Avg |
|---|---|---|---|---|---|---|---|---|---|---|---|---|---|
| 16, 2, 1024 | 16.23 | **53.0** | 41.3 | **79.8** | 65.0 | 32.6 | 37.8 | **66.0** | 32.2 | 55.8 | 53.9 | **29.1** | 49.7 |
| 64, 8, 256 | **16.01** | 51.0 | 41.4 | 79.2 | **65.5** | 31.6 | **38.4** | **66.0** | 31.5 | 60.2 | 57.5 | 25.7 | 49.8 |
| 256, 32, 64 | 16.13 | 51.2 | **41.5** | 78.9 | 65.3 | **34.2** | **38.4** | 63.0 | **32.4** | **60.6** | 59.5 | 28.1 | **50.3** |
| Dense, iso-FLOPs | 19.90 | 48.9 | 41.4 | 74.9 | 62.2 | 30.2 | 32.6 | 62.0 | 31.6 | 61.7 | 53.2 | 27.1 | 47.8 |
| Dense, iso-params | 15.46 | 52.1 | 41.5 | 78.9 | 65.3 | 34.0 | 39.2 | 69.0 | 32.2 | 58.5 | 59.3 | 28.8 | 50.8 |

(b) **1.4B params, 50B tokens, 8/128 activated**

| $(E,K,n)$ | PPL | Wino | SIQA | SciQ | PIQA | OBQA | HS | COPA | CSQA | BoolQ | ArcE | ArcC | Avg |
|---|---|---|---|---|---|---|---|---|---|---|---|---|---|
| 32, 2, 1024 | 13.38 | **52.2** | **41.7** | 81.7 | 69.2 | 33.6 | 44.3 | 64.0 | 33.5 | **61.1** | 60.9 | 29.8 | 52.0 |
| 128, 8, 256 | **13.32** | 51.8 | **41.7** | 81.5 | **69.3** | 32.4 | **45.3** | 68.0 | **34.5** | 56.6 | **63.2** | 28.4 | 52.1 |
| 512, 32, 64 | 13.50 | 52.5 | 41.2 | **82.9** | 68.9 | **34.4** | 44.7 | **69.0** | 33.6 | 58.7 | 62.6 | **30.1** | **52.6** |
| Dense, iso-FLOPs | 17.90 | 52.2 | 41.0 | 79.2 | 63.4 | 31.0 | 34.7 | 61.0 | 30.5 | 60.3 | 51.8 | 25.1 | 48.2 |
| Dense, iso-params | 12.74 | 52.2 | 42.6 | 83.3 | 70.1 | 34.8 | 46.8 | 67.0 | 35.1 | 61.7 | 63.5 | 31.8 | 53.5 |

## E.2    ABLATION STUDY ON DIFFERENT ROUNDING SUBROUTINES FOR TOKEN ROUNDING

We conduct ablation studies to study the effect of the different routing subroutines on the trained MoE by TR. We compare token rounding with nearest rounding ("NR") with per-expert token counts with other rounding methods. Specifically, we compare against stochastic rounding with per-expert token count ("SR"), always round up ("UP"), and always round down ("DOWN"). The results are shown in Table 6 and we find that our token rounding algorithm in general is robust to the specific rounding subroutines.

Following Algorithm 4, for expert $e$, we denote the expert frequency from the TC sorting as $f_e$, and the last $M_{\text{tile}}$-divisible expert frequency as $\lfloor f_e \rfloor_{M_{\text{tile}}}$, and the next $M_{\text{tile}}$-divisible expert frequency as $\lceil f_e \rceil_{M_{\text{tile}}}$. We also denote the expert scores from TC sorting as $s_e$, the expert scores from the selected tokens in $\pi_e[: \lfloor f_e \rfloor_{M_{\text{tile}}}]$ as $\lfloor s_e \rfloor_{M_{\text{tile}}}$ and the scores for $\pi_e[: \lceil f_e \rceil_{M_{\text{tile}}}]$ as $\lceil s_e \rceil_{M_{\text{tile}}}$. We note that all rounding algorithms only make a *binary decision* between discarding TC tokens and padding EC tokens for each expert. Following are simple heuristics to perform rounding:

- **NR-f: nearest rounding to $M_{\text{tile}}$-multiples via expert frequency**: We pad EC tokens if $\lceil f_e \rceil_{M_{\text{tile}}} - f_e < f_e - \lfloor f_e \rfloor_{M_{\text{tile}}}$. "NR-f" is our default choice of token rounding and we use it for Table 1, 7, 8, and Figures 3 and 7.

---

**Algorithm 6** Balanced rounding to $M_{\text{tile}}$-multiples via expert frequency ("Balance-f" in Table 6).
This algorithm satisfies $\max_{e \in [E]} \left| \lceil f_e \rceil_{M_{\text{tile}}} - f_e \right| \leq M_{\text{tile}}/2$ and $\left| \sum_{e=1}^{E} \lceil f_e \rceil_{M_{\text{tile}}} - \sum_{e=1}^{E} f_e \right| \leq M_{\text{tile}}/2$

---

**Input** : $f^{\text{TC}} = \{f_e\}_{e \in [E]}$ as a list of expert frequency with TC top-$K$ routing, $\{\lceil f_e \rceil_{M_{\text{tile}}}\}_{e \in [E]}$ as a list of expert frequency with TC top-$K$
routing and with potential EC padding to ensure each expert receives a multiple of $M_{\text{tile}}$ tokens, $\{\lfloor f_e \rfloor_{M_{\text{tile}}}\}_{e \in [E]}$ as a list of expert
frequency with TC top-$K$ routing and with potential token dropping to ensure each expert receives a multiple of $M_{\text{tile}}$ tokens; We should
ensure $\max_{e \in [E]} \left( \lceil f_e \rceil_{M_{\text{tile}}} - \lfloor f_e \rfloor_{M_{\text{tile}}} \right) \leq M_{\text{tile}}$.

**Output** : $f^{\text{TR}} = \{\lceil f_e \rceil_{M_{\text{tile}}}\}_{e \in [E]}$ as a list of expert frequency that ensures each expert receives a multiple of $M_{\text{tile}}$ tokens

// an accumulator that ensures the preservation of total expert frequency

$z \leftarrow 0$;

**for** $e \in [E]$ **do**
    // calculate the residual error of both rounding choice
    $r_e^{\text{up}} \leftarrow \lceil f_e \rceil_{M_{\text{tile}}} - f_e$;
    $r_e^{\text{down}} \leftarrow \lfloor f_e \rfloor_{M_{\text{tile}}} - f_e$;
    **if** $|r_e^{up} + z| < |r_e^{down} + z|$ **then**
        // choose to pad with EC tokens
        $\lceil f_e \rceil_{M_{\text{tile}}} \leftarrow \lceil f_e \rceil_{M_{\text{tile}}}$;
        $z \leftarrow z + r_e^{\text{up}}$;
    **else**
        // choose to discard TC tokens
        $\lceil f_e \rceil_{M_{\text{tile}}} \leftarrow \lfloor f_e \rfloor_{M_{\text{tile}}}$;
        $z \leftarrow z + r_e^{\text{down}}$;

---

- **SR-f: stochastic rounding to $M_{\text{tile}}$-multiples via expert frequency**: We sample from
  $\text{Bernoulli}\left( \dfrac{f_e - \lfloor f_e \rfloor_{M_{\text{tile}}}}{M_{\text{tile}}} \right)$ distribution for deciding whether to pad EC tokens for expert $e$.

- **NR-s: nearest rounding to $M_{\text{tile}}$-multiples via expert scores**: We sample from the following
  distribution for deciding whether to pad EC tokens for expert $e$:

$$\text{Bernoulli}\left( \frac{\sum_t s_{e,t} - \sum_t \lfloor s_{e,t} \rfloor_{M_{\text{tile}}}}{\sum_t \lceil s_{e,t} \rceil_{M_{\text{tile}}} - \sum_t \lfloor s_{e,t} \rfloor_{M_{\text{tile}}}} \right) \tag{13}$$

- **Balance-f: balanced rounding to $M_{\text{tile}}$-multiples via expert frequency**: The Balance
  algorithm (Dwivedi & Mackey, 2024; Lu et al., 2022; Cooper et al., 2023) can be adapted
  to ensure the total number of routed tokens to all experts after tile-rounding is preserved
  regardless of the number of experts $E$. Algorithm 6 is such an example that ensures

$$\max_{e \in [E]} \left| \lceil f_e \rceil_{M_{\text{tile}}} - f_e \right| \leq M_{\text{tile}}/2, \qquad \left| \sum_{e=1}^{E} \lceil f_e \rceil_{M_{\text{tile}}} - \sum_{e=1}^{E} f_e \right| \leq M_{\text{tile}}/2 \tag{14}$$

  where the other rounding subroutine will have an expected deviation of $O\left( M_{\text{tile}} \sqrt{E} \right)$ for
  $\sum_{e=1}^{E} \lceil f_e \rceil_{M_{\text{tile}}}$.

- **UP: always round up expert frequency as $\lceil f_e \rceil_{M_{\text{tile}}}$**: We always pad EC tokens chosen
  in the second step of sorting in Algorithm 4. This gives a model TFLOPS lower-bound for
  Figure 7.

- **DOWN: always round down expert frequency as $\lfloor f_e \rfloor_{M_{\text{tile}}}$**: We always discard TC top-
  $K$ tokens chosen in the first step of sorting in Algorithm 4. This gives a model TFLOPS
  upper-bound for Figure 7.

**Always discarding TC tokens ("DOWN").** "DOWN" is a baseline in which we always drop the last
TC tile if the expert frequency is not $M_{\text{tile}}$-divisible. This idea is similar to the idea of *token dropping* in
expert parallelism where the expert will sort and drop the token with the lowest scores when it receives
too many tokens (Fedus et al., 2022). We note that "DOWN" produces the shortest MoE kernel runtime

Table 6: Evaluation of token rounding algorithms equipped with different round_and_sparsify subroutines in Algorithm 4. "PPL" refers to the validation perplexity at the end of training. "Avg" is the mean accuracy across the 11 downstream tasks.

(a) **0.5B params, 40B tokens, 2/64 activated** ($\bar{T}_e = 512$, $M_{\text{tile}} = 128$)

| Method | PPL | Wino | SIQA | SciQ | PIQA | OBQA | HS | COPA | CSQA | BoolQ | ArcE | ArcC | Avg |
|---|---|---|---|---|---|---|---|---|---|---|---|---|---|
| TR (NR-f) | 15.92 | 51.4 | 41.6 | 78.4 | 65.4 | 31.6 | 38.1 | 65.0 | 31.0 | 61.1 | 57.4 | 29.1 | 50.0 |
| TR (SR-f) | 15.93 | 50.8 | 40.9 | 77.4 | **66.9** | **33.0** | **38.4** | 64.0 | 31.1 | 60.7 | 55.8 | 28.1 | 49.7 |
| TR (NR-s) | 15.91 | 51.3 | 40.9 | **80.3** | 65.4 | 30.8 | 37.7 | 67.0 | 31.0 | 61.6 | 55.4 | 28.4 | 50.0 |
| TR (Balance-f) | 15.93 | **51.9** | **41.8** | 78.8 | 65.9 | 32.6 | **38.4** | 66.0 | 31.6 | 60.3 | 56.8 | 27.1 | 50.1 |
| TR (UP) | **15.89** | 50.5 | 40.9 | 78.6 | 64.5 | 32.2 | 38.2 | **68.0** | 29.9 | 55.2 | 54.2 | 30.1 | 49.3 |
| TR (DOWN) | 16.10 | 51.1 | 41.4 | 78.7 | 64.9 | 31.6 | 38.0 | 62.0 | **32.8** | **61.9** | **58.9** | **30.8** | **50.2** |
| TC top-$K$ | 15.94 | 51.0 | 41.9 | 78.5 | 64.8 | 33.0 | 38.1 | 67.0 | 30.8 | 54.7 | 55.8 | 30.1 | 49.6 |

(b) **1.8B params, 40B tokens, 8/256 activated** ($\bar{T}_e = 512$, $M_{\text{tile}} = 128$)

| Method | PPL | Wino | SIQA | SciQ | PIQA | OBQA | HS | COPA | CSQA | BoolQ | ArcE | ArcC | Avg |
|---|---|---|---|---|---|---|---|---|---|---|---|---|---|
| TR (NR-f) | 13.10 | 53.4 | 42.1 | 81.7 | 69.6 | 35.2 | 45.3 | **70.0** | 33.2 | **61.4** | 63.0 | 33.4 | **53.5** |
| TR (SR-f) | 13.08 | 52.7 | 41.6 | 82.6 | 69.4 | 34.4 | 45.6 | **70.0** | 33.0 | 59.1 | 62.5 | **34.8** | 53.2 |
| TR (NR-s) | 13.09 | 54.1 | **42.3** | **82.8** | 69.3 | 33.8 | **45.7** | **70.0** | 34.1 | 59.0 | **64.6** | 32.4 | **53.5** |
| TR (Balance-f) | 13.08 | 52.5 | 42.0 | 82.7 | **70.0** | 33.2 | 45.3 | 68.0 | **34.6** | 59.4 | 63.3 | 33.4 | 53.1 |
| TR (UP) | **13.07** | 50.4 | 41.7 | 81.4 | 68.4 | **37.2** | 45.4 | 69.0 | 31.9 | 51.7 | 62.2 | 33.4 | 52.1 |
| TR (DOWN) | 13.19 | **55.4** | 41.6 | 82.2 | 68.6 | 34.8 | 45.0 | 69.0 | 34.0 | 54.4 | 63.5 | 31.4 | 52.7 |
| TC top-$K$ | 13.12 | 50.1 | 42.9 | 81.3 | 69.8 | 33.8 | 45.2 | 71.0 | 34.1 | 56.7 | 64.6 | 31.1 | 52.8 |

for any rounding algorithm. However, in Table 6, we observe that "DOWN" yields a much higher validation perplexity than "NR-f", "SR-f" and "NR-s". Although we can expect a shorter MoE kernel runtime by always discarding TC tokens, such quality degradation might not be acceptable in practice.

**Always padding EC tokens ("UP").** "UP" is a baseline in which we always pad extra EC tokens to the last TC tile if the expert frequency is not $M_{\text{tile}}$-divisible. Contrary to "DOWN", "UP" produces the longest MoE kernel runtime for any rounding algorithm. In Table 6, we find that "UP" often produces lower validation perplexity, but the average downstream task accuracy is not necessarily higher than other rounding algorithms. Given the longer MoE kernel runtime but not necessarily better trained MoE quality, we do not recommend the usage of always rounding up. *We speculate this is due to the train-test gap between TC and EC routing and "UP" reinforces the bias towards EC more strongly than other TR algorithms.*

For a balance between training efficiency and trained MoE quality, neither always discarding TC tokens nor padding EC tokens is the right solution. In Table 1, we pick "NR-f" as the round_and_sparsify subroutine for TR's main experiments.

### E.3 Ablation study on the effects of microbatch size $T$ and tile size $M_{\text{tile}}$

**Effect of microbatch size $T$.** Since the token rounding is applied on the microbatch level, the choice of microbatch size $T$ will result in different qualitative results for TR. Note that this also holds true for EC routing. For example, EC over sequence will result in different model quality than EC over a text segment. In Table 7, we vary the microbatch size while keeping the minibatch size (consumed tokens per optimization step) constant.

We find that TR will preserve its trained MoE quality when $\bar{T}_e / M_{\text{tile}} \geq 2$, but if $\bar{T}_e / M_{\text{tile}} = 1$ (the last row in both subtables), there is a noticeable quality degradation for both validation perplexity and downstream task performance. However, the trained MoE quality with $\bar{T}_e / M_{\text{tile}} = 1$ is still better than training with EC and finetuning with TC top-$K$ routing.

**Effect of the tile quantization size $M_{\text{tile}}$.** Similarly in Table 8, we can find that TR is generally robust w.r.t. $M_{\text{tile}}$ when $\bar{T}_e / M_{\text{tile}} \geq 2$, and when $\bar{T}_e / M_{\text{tile}} = 1$ there is a noticeable degradation but the overall result is still better than EC baseline.

Table 7: Evaluation of token rounding algorithms when we vary microbatch size $T$ to change average number of tokens per expert ($\bar{T}_e$). For each trial, we vary the microbatch size from 4 ($\bar{T}_e = 512$) to 1 ($\bar{T}_e = 128$) and keep minibatch size constant. The $M_{\text{tile}}$ is always kept as 128. "PPL" refers to the validation perplexity at the end of training. "Avg" is the mean accuracy across the 11 downstream tasks.

(a) **0.5B params, 40B tokens, 2/64 activated** ($M_{\text{tile}} = 128$)

| Method | PPL | Wino | SIQA | SciQ | PIQA | OBQA | HS | COPA | CSQA | BoolQ | ArcE | ArcC | Avg |
|---|---|---|---|---|---|---|---|---|---|---|---|---|---|
| TR ($\bar{T}_e = 1024$) | **15.91** | 52.9 | 41.0 | 80.1 | 65.1 | 31.0 | 37.9 | 63.0 | 32.3 | 59.3 | 54.9 | 28.1 | 49.6 |
| **TR** ($\bar{T}_e = 512$) | 15.92 | 51.4 | 41.6 | 78.4 | 65.4 | 31.6 | 38.1 | 65.0 | 31.0 | 61.1 | 57.4 | 29.1 | 50.0 |
| TR ($\bar{T}_e = 256$) | 15.98 | 52.2 | 41.4 | 77.7 | 66.1 | 32.2 | 37.9 | 66.0 | 31.0 | 59.6 | 57.2 | 30.1 | 50.1 |
| TR ($\bar{T}_e = 128$) | 16.11 | 51.7 | 41.7 | 77.9 | 66.1 | 30.8 | 37.7 | 67.0 | 31.9 | 61.2 | 54.7 | 29.1 | 50.0 |
| TC top-$K$ | 15.94 | 51.0 | 41.9 | 78.5 | 64.8 | 33.0 | 38.1 | 67.0 | 30.8 | 54.7 | 55.8 | 30.1 | 49.6 |
| EC (ft TC router) | 16.98 | 50.0 | 41.7 | 79.7 | 64.9 | 31.6 | 36.8 | 63.0 | 32.1 | 60.7 | 54.6 | 27.4 | 49.3 |

(b) **1.8B params, 40B tokens, 8/256 activated** ($M_{\text{tile}} = 128$)

| Method | PPL | Wino | SIQA | SciQ | PIQA | OBQA | HS | COPA | CSQA | BoolQ | ArcE | ArcC | Avg |
|---|---|---|---|---|---|---|---|---|---|---|---|---|---|
| TR ($\bar{T}_e = 1024$) | **13.08** | 51.5 | 42.0 | 81.7 | 68.9 | 34.8 | 45.7 | 72.0 | 32.6 | 59.5 | 61.4 | 32.1 | 52.9 |
| **TR** ($\bar{T}_e = 512$) | 13.10 | 53.4 | 42.1 | 81.7 | 69.6 | 35.2 | 45.3 | 70.0 | 33.2 | 61.4 | 63.0 | 33.4 | 53.5 |
| TR ($\bar{T}_e = 256$) | 13.12 | 51.9 | 41.2 | 81.8 | 69.7 | 33.6 | 45.2 | 73.0 | 34.2 | 56.9 | 63.2 | 34.1 | 53.2 |
| TR ($\bar{T}_e = 128$) | 13.55 | 51.9 | 41.5 | 82.0 | 69.2 | 32.8 | 44.4 | 69.0 | 34.4 | 59.8 | 64.0 | 30.4 | 52.7 |
| TC top-$K$ | 13.12 | 50.1 | 42.9 | 81.3 | 69.8 | 33.8 | 45.2 | 71.0 | 34.1 | 56.7 | 64.6 | 31.1 | 52.8 |
| EC (ft TC router) | 15.01 | 52.7 | 41.1 | 79.6 | 66.9 | 30.6 | 40.2 | 66.0 | 31.9 | 60.5 | 57.2 | 30.8 | 50.7 |

Table 8: Evaluation of token rounding algorithms when we vary the size of tile $M_{\text{tile}}$ for token rounding. "PPL" refers to the validation perplexity at the end of training. "Avg" is the mean accuracy across the 11 downstream tasks.

(a) **0.5B params, 40B tokens, 2/64 activated** ($\bar{T}_e = 512$)

| Method | PPL | Wino | SIQA | SciQ | PIQA | OBQA | HS | COPA | CSQA | BoolQ | ArcE | ArcC | Avg |
|---|---|---|---|---|---|---|---|---|---|---|---|---|---|
| TR ($M_{\text{tile}} = 64$) | **15.90** | 51.3 | 41.7 | 78.1 | 65.6 | 31.4 | 37.9 | 67.0 | 32.4 | 59.8 | 57.2 | 28.8 | 50.1 |
| **TR** ($M_{\text{tile}} = 128$) | 15.92 | 51.4 | 41.6 | 78.4 | 65.4 | 31.6 | 38.1 | 65.0 | 31.0 | 61.1 | 57.4 | 29.1 | 50.0 |
| TR ($M_{\text{tile}} = 256$) | 16.00 | 51.7 | 41.4 | 78.7 | 66.3 | 32.4 | 37.7 | 67.0 | 31.3 | 60.1 | 58.2 | 29.1 | 50.4 |
| TR ($M_{\text{tile}} = 512$) | 16.17 | 52.5 | 41.2 | 80.2 | 65.2 | 32.0 | 37.9 | 62.0 | 31.0 | 59.4 | 57.2 | 30.4 | 49.9 |
| TC top-$K$ | 15.94 | 51.0 | 41.9 | 78.5 | 64.8 | 33.0 | 38.1 | 67.0 | 30.8 | 54.7 | 55.8 | 30.1 | 49.6 |
| EC (ft TC router) | 16.98 | 50.0 | 41.7 | 79.7 | 64.9 | 31.6 | 36.8 | 63.0 | 32.1 | 60.7 | 54.6 | 27.4 | 49.3 |

(b) **1.8B params, 40B tokens, 8/256 activated** ($\bar{T}_e = 512$)

| Method | PPL | Wino | SIQA | SciQ | PIQA | OBQA | HS | COPA | CSQA | BoolQ | ArcE | ArcC | Avg |
|---|---|---|---|---|---|---|---|---|---|---|---|---|---|
| TR ($M_{\text{tile}} = 64$) | **13.07** | 52.3 | 42.9 | 82.7 | 69.4 | 35.4 | 45.6 | 70.0 | 32.4 | 56.6 | 64.4 | 31.4 | 53.0 |
| **TR** ($M_{\text{tile}} = 128$) | 13.10 | 53.4 | 42.1 | 81.7 | 69.6 | 35.2 | 45.3 | 70.0 | 33.2 | 61.4 | 63.0 | 33.4 | 53.5 |
| TR ($M_{\text{tile}} = 256$) | 13.13 | 52.0 | 41.6 | 82.1 | 69.2 | 35.4 | 45.3 | 69.0 | 34.2 | 58.0 | 65.6 | 32.1 | 53.1 |
| TR ($M_{\text{tile}} = 512$) | 13.56 | 53.0 | 41.8 | 81.2 | 68.4 | 34.0 | 44.2 | 68.0 | 33.3 | 58.1 | 59.5 | 30.1 | 52.0 |
| TC top-$K$ | 13.12 | 50.1 | 42.9 | 81.3 | 69.8 | 33.8 | 45.2 | 71.0 | 34.1 | 56.7 | 64.6 | 31.1 | 52.8 |
| EC (ft TC router) | 15.01 | 52.7 | 41.1 | 79.6 | 66.9 | 30.6 | 40.2 | 66.0 | 31.9 | 60.5 | 57.2 | 30.8 | 50.7 |

# F ACTIVATION MEMORY AND TRAINING THROUGHPUT BENCHMARK CONFIGURATIONS

The configurations of Figure 5, 6a, and 6b are included in Table 9.

The configurations for the 4 subfigures in Figure 7 are listed below. Notice that we consistently use $M_{\text{tile}}$ as 128 when we benchmark the TR's speed.

- **Top-left 2 subfigures**: We use $(T, d, n, K) = (16384, 1536, 256, 8)$ and we vary $E$ from 64 to 512.

- **Top-right 2 subfigures**: We use $(T, d, n, K) = (16384, 1536, 1024, 2)$ and we vary $E$ from 16 to 128.

- **Bottom-left 2 subfigures**: We use $(T, d, n, K) = (16384, 4096, 512, 8)$ and we vary $E$ from 64 to 512.

Table 9: Benchmark configurations for activation memory footprint and compute throughput.

(a) Benchmark configurations used by Figure 5 and 6a.

| Model Size | $T$ | $d$ | $n$ | $E$ | $K$ |
|---|---|---|---|---|---|
| 1.4B | 40960 | 768 | 256 | 128 | 8 |
| | 40960 | 768 | 512 | 64 | 4 |
| | 40960 | 768 | 1024 | 32 | 2 |
| 7B | 24576 | 1536 | 256 | 128 | 8 |
| | 24576 | 1536 | 512 | 64 | 4 |
| | 24576 | 1536 | 1024 | 32 | 2 |
| 30B | 32768 | 4096 | 256 | 256 | 16 |
| | 32768 | 4096 | 512 | 128 | 8 |
| | 32768 | 4096 | 1024 | 64 | 4 |
| 120B | 32768 | 4096 | 512 | 256 | 16 |
| | 32768 | 4096 | 1024 | 128 | 8 |
| | 32768 | 4096 | 2048 | 64 | 4 |

(b) Benchmark configurations used by Figure 6b

| Model Size | $T$ | $d$ | $n$ | $E$ | $K$ |
|---|---|---|---|---|---|
| 1.4B | 131072 | 768 | 256 | 128 | 8 |
| | 131072 | 768 | 512 | 64 | 4 |
| | 131072 | 768 | 1024 | 32 | 2 |
| 7B | 81920 | 1536 | 256 | 128 | 8 |
| | 81920 | 1536 | 512 | 64 | 4 |
| | 81920 | 1536 | 1024 | 32 | 2 |
| 30B | 32768 | 4096 | 256 | 256 | 16 |
| | 32768 | 4096 | 512 | 128 | 8 |
| | 32768 | 4096 | 1024 | 64 | 4 |
| 120B | 32768 | 4096 | 512 | 256 | 16 |
| | 32768 | 4096 | 1024 | 128 | 8 |
| | 32768 | 4096 | 2048 | 64 | 4 |

- **Bottom-right 2 subfigures**: We use $(T,d,n,K)=(16384,4096,1024,4)$ and we vary $E$ from 32 to 256.

## G  HYPERPARAMETER DETAILS FOR LM TRAINING

We use the OLMoE codebase (Muennighoff et al., 2025) and its downstream tasks in the default configuration[12] except for MMLU: WinoGrande ("wino") (Sakaguchi et al., 2020), Social IQA ("SIQA") (Sap et al., 2019), SciQ (Johannes Welbl, 2017), PIQA (Bisk et al., 2020), OpenBookQA ("OBQA") (Mihaylov et al., 2018), HellaSwag ("HS") (Zellers et al., 2019), COPA (Roemmele et al., 2011), CommonsenseQA ("CSQA") (Talmor et al., 2019), BoolQ (Clark et al., 2019), Arc-Easy and Arc-Challenge ("ArcE" and "ArcC") (Clark et al., 2018) datasets. We use a deduplicated version of FineWeb-Edu (Ben Allal et al., 2024)[13] for pretraining corpus, and train all models with context length of 4096 tokens.

We always use MoE with SwiGLU for the MoE layers and we use an auxiliary load balancing loss (Shazeer et al., 2017) with coefficient 0.01 but we do not use the router Z loss (Zoph et al., 2022). Our attention block architecture is identical to OLMoE's attention block. We always tie the weight of the LM head with the weight of the token embedding matrix.

Table 10: Common configurations for MoE pretraining experiment

| Config name in Tables 1 and 6 | # layers | # attn heads | $d$ | $n$ | $E$ | $K$ | minibatch token size | LR | WD | LR scheduler |
|---|---|---|---|---|---|---|---|---|---|---|
| 0.5B params, 40B tokens, $\rho=2/64$ | 12 | 12 | 768 | 256 | 64 | 2 | 1M | 6e-4 | 0.01 | cos w/. warmup (10% steps) |
| 1.8B params, 40B tokens, $\rho=8/256$ | 12 | 12 | 768 | 256 | 256 | 8 | 1M | 6e-4 | 0.01 | cos w/. warmup (10% steps) |
| 1.4B params, 100B tokens, $\rho=2/128$ | 18 | 12 | 768 | 256 | 128 | 2 | 2M | 4e-4 | 0.01 | cos w/. warmup (10% steps) |

For all EC with finetuned TC router experiments in Table 1, we use an additional 4B tokens and we only finetune the router weights with TC top-$K$ routing (all other parameters are frozen). We always use a learning rate of 2e-4, weight decay of 0.01 and cosine learning rate scheduler with 10% warmup steps. The number of tokens per minibatch during finetuning is 1M. We disable auxiliary load balancing loss during TC finetuning.

For all EC with auxiliary router experiments, we use a 2-layer MLP (each linear layer has size $E \times E$ with SiLU activation) which takes as input the raw router logits and makes $E$ independent binary predictions for all experts. We compute the averaged binary cross entropy loss over $E$ labels using the multi-label prediction loss, and scale the loss by 0.01. During the evaluation, we will let EC router

---

[12] https://github.com/allenai/OLMoE/blob/357454f4f647385839c0ff6b99a688dc7cd9c13f/configs/OLMoE-1B-7B-0924.yml

[13] https://huggingface.co/datasets/HuggingFaceTB/smollm-corpus

compute the raw logits and raw scores and let the auxiliary router mask the token-expert pair with its own confidence score.

We implement "TC (token drop)" by discarding tokens selected from the TC top-$K$ sorting, or always round down.

