# OpenReview forum: "SonicMoE: Accelerating MoE with IO and Tile-aware Optimizations"
_ICLR.cc/2026/Conference — ICLR 2026 Poster_

### Official Review · Reviewer_Y7dy · 2025-10-27

**Soundness:** 4
**Presentation:** 3
**Contribution:** 3
**Rating:** 8
**Confidence:** 4

**Summary:**

This paper, SNaX, presents three improvements to current MoE models to allow them to better utilize modern GPU hardware. Specifically, they propose a new implementation that reduces the memory used to store activations during training and allow this activation size to be independent of the expert granularity. Secondly, they improve throughput by overlapping IO with computation. Finally, they propose to constrain the routing strategy such that the number of tokens routed to each expert is a multiple of the tile size of GPUs, reducing the amount of wasted computation resulted from padding. Empirical results show that the propose implementation and algorithm is much more efficient in terms of throughput and memory usage while maintaining performance. It also fosters the development of high-granularity MoE models since it provides better support for such models compared to existing open-source solutions.

**Strengths:**

- The paper is well-written, the motivation is clear, and the description of the proposed algorithm is easy to understand.
- The experimental results are convincing and the proposed method provides considerable efficiency improvements. The proposed method, which will be open-source as described in Line 107, will be useful for the community at large since it will greatly lower the cost of training large MoE models. Moreover, high-granularity MoE models hold great promise and is the focus of many research works. The method proposed by this paper will be helpful for many researchers.

**Weaknesses:**

- It would be helpful to see efficiency tests on more hardware other than H100.
- It seems that the focus of this paper lies in training efficiency. It is currently unclear whether parts of the improvements can be used to improve inference efficiency. Nonetheless, a discussion on inference efficiency seems to be important.

**Questions:**

- Figure 5 and Section 6.2: Floating-point operations per second is usually abbreviated as FLOPS instead of FLOPs.
- I think the citation in Line 264 should be moved to the first reference of Ping-Pong in Line 256. Moreover, "Ping-Pong" and "pingpong" both appears in the paper. The authors, should unify the capitalization for this term.
- Figure 6 is never referred to in the paper.
- Can this method be used for MoE models without a top-K function, such as ReMoE [1]?

[1] ReMoE: Fully Differentiable Mixture-of-Experts with ReLU Routing

---

> ### Author Response · Authors · 2025-11-21
> **Response to reviewer Y7dy (1/1)**
>
> ## To Reviewer Y7dy
>
> We thank reviewer Y7dy for appreciation of our clear motivation and writing, convincing experiment results, and the future impact of our method after the open-source release for many researchers working on MoEs. Below we will address reviewer Y7dy's questions and concerns.
>
> ## W1 - efficiency on hardware other than H100
> > It would be helpful to see efficiency tests on more hardware other than H100.
>
> Thanks for the suggestion! We are currently working on kernels for Blackwell GPUs and will update our benchmark results for the forward pass soon.
>
>
> ## W2 - inference efficiency
>
> > It seems that the focus of this paper lies in training efficiency. It is currently unclear whether parts of the improvements can be used to improve inference efficiency. Nonetheless, a discussion on inference efficiency seems to be important.
>
> Thanks for positioning SNaX's efficiency in a greater scope. In this paper, we are primarily focusing on the training efficiency of MoEs. Our kernels in forward pass can be adapted (after disabling activation caching for backward to GMEM) for inference efficiency. **We note that SNaX's main benefits come from the large-batch inference instead of small or single-batch inference. In addition, the speedup mostly arises from using our kernels for prefill instead of decode.** This is because SNaX is designed for overlapping the IO with MMA compute, and to achieve a meaningful IO overlap, the computational workloads need to provide sufficiently large MMA (often via a large batch). Such requirement is usually fulfilled during training or in the prefill phase during inference.
>
>
> ## Q1 - Notation on FLOP per second
>
> > Figure 5 and Section 6.2: Floating-point operations per second is usually abbreviated as FLOPS instead of FLOPs.
>
> Thanks for the suggestion! We will update the figure and the main text to follow the correct convention.
>
> ## Q2 - Move citation of Ping-Pong and unify capitalization
>
> > I think the citation in Line 264 should be moved to the first reference of Ping-Pong in Line 256. Moreover, "Ping-Pong" and "pingpong" both appears in the paper. The authors, should unify the capitalization for this term.
>
> Thanks for the suggestion! We will unify our capitalization to "Ping-Pong" and move the citation.
>
> ## Q3 - reference of Figure 6
>
> > Figure 6 is never referred to in the paper.
>
> Thanks for mentioning the Figure 6! In Figure 6 we profile the breakdown in milliseconds for each kernel, and Figure 6 explains the SNaX's speedup over existing MoE kernel designs. We will annotate the TFLOPs and TB/s for each kernel and add more detailed analysis to illustrate SNaX's speedup. In Section 4 we reformulate the computation of $dZ$, and fuse the calculations of $dZ$, $dS$ and $Y_4$ into a single kernel, resulting in a heavy epilogue that can be further overlapped with MMA using Ping-Pong. **The Figure 6 shows that this heavy epilogue fusion results in a major speedup over alternative designs including ScatterMoE since SNaX's $dH$ kernel (0.47ms, 333 TFLOPS) takes the same time as ScatterMoE's down-proj act (0.43ms, 364 TFLOPs), $dS$ (0.24ms), and dSwiGLU (0.33ms) combined together (0.99ms) during 7B MoE training in Figure 6.**
>
> We are also working on adding more baselines (DeepGEMM and Megatron-LM's GroupedMLP) and provide detailed annotations and analysis for Figure 6 to illustrate the speedup of SNaX.
>
> ## Q4 - SNaX's application besides TC Top-K
>
> > Can this method be used for MoE models without a top-K function, such as ReMoE?
>
> Thanks for bringing SNaX's custom router interface to attention! The MoE routing is completely decoupled with the MoE computation kernel in SNaX. The main modification of ReMoE from top-K is ReLU-based routing and sparsity-regularization loss. In this case, we only need to implement a custom ReLU router, and use the custom router inteface in SNaX that supports arbitrary routing inputs. The additional gradient signal from sparsity-regularization loss on the router weight will be summed up by PyTorch AD engine. **Such decoupling allows future researchers to iterate faster on new MoE routing techniques without impacting training efficiency**.
>
> This is similar to the case for our token rounding routing method. Our SNaX kernel will output $dS$ as normal, and we partially rely on PyTorch AD engine for computing the router input and weight gradient based on the new $dS$. We have verified the numerical accuracy of all weight and input gradients with a reference PyTorch implementation when we use SNaX custom routing interface.

---

> ### Comment · Reviewer_Y7dy · 2025-11-24
> **Response to Rebuttal**
>
> I appreciate the authors response to my review. I have decided to maintain my score, which is quite high already. I look forward to seeing the revised version of the paper.

---

### Official Review · Reviewer_tULt · 2025-10-31

**Soundness:** 3
**Presentation:** 3
**Contribution:** 2
**Rating:** 4
**Confidence:** 3

**Summary:**

This paper addresses the memory bottlenecks and hardware inefficiencies posed by the trend towards increasingly fine-grained and sparse Mixture of Experts (MoE) models. It proposes SNaX, a system and algorithm co-design solution. SNaX tackles these challenges via three core contributions: (1) A memory-efficient MoE algorithm that minimizes activation memory by modifying the backward pass computation graph; (2) An IO-aware GPU kernel design that overlaps memory IO latency with computation ; and (3) A novel 'Token Rounding' routing method designed to reduce wasted compute resulting from tile quantization. Experimental results show that SNaX significantly reduces memory usage (e.g., a 45% reduction for a fine-grained 7B MoE model ) while substantially improving compute throughput (a 1.87x improvement on Hopper GPUs ).

**Strengths:**

● The paper astutely points out that as MoE models trend towards being more fine-grained and sparse, the training bottleneck is shifting from computation to memory bandwidth.
● Consistent performance improvements over baselines like ScatterMoE and MoMoE across various model scales (1.4B to 120B) strongly demonstrate the practical value of the proposed method.
● 'Token Rounding' is a novel and clever idea that addresses the often-overlooked problem of wasted computation in GEMM operations due to padding, which the paper refers to as 'tile quantization'.

**Weaknesses:**

● The paper states in Section 6.3 that Token Rounding (TR) is used during training, but it switches back to Token Choice (TC) for evaluation. The authors also admit that "Token rounding is not a token-choice routing method which creates difficulty for autoregressive inference." This is a major concern. The authors do not clearly explain the root cause of this 'difficulty'.
● Furthermore, this training-inference mismatch constitutes a theoretical weakness. Although the results in Table 1 show acceptable PPL and Avg accuracy, it is unclear if this mismatch could negatively impact other aspects of the model, such as generation diversity or the prevalence of 'hallucinations'.
● Moreover, the paper does not discuss the differences in loss convergence between the two methods (TR vs. TC) during the training phase.
● The paper highlights 'Ping-Pong' scheduling and asynchronous TMA as key to its IO-aware kernel design. While their application in SNaX is effective, these scheduling techniques themselves are not new. The authors should more clearly articulate that their core contribution lies in the specific application and adaptation of these techniques to the heavy-epilogue nature of MoE kernels, rather than claiming novelty for the techniques themselves.
● The paper fails to clarify certain technical terms. For instance, what is the "state-of-the-art BF16 MoE kernel" that SNaX is compared against? Additionally, what does the "K dimension" (mentioned in Section 2.1) refer to in the context of GEMM?

**Questions:**

The paper avoids caching Y_2 by reordering the computation graph. Does this complex backward pass introduce any numerical instabilities, particularly when training in mixed precision (BF16)?

---

> ### Author Response · Authors · 2025-11-22
> **Response to reviewer tULt (1/2)**
>
> We thank the reviewer for appreciation on our clear and valuable motivation on the efficiency bottlenecks of training highly sparse MoEs that computational workloads are shifting into memory bandwidth, consistent throughput improvements across model scales, and clear and novel idea of token rounding. Below we will address reviewer tULt's questions and concerns.
>
> ## W1, W2 - training-inference consistency
>
> > The paper states in Section 6.3 that Token Rounding (TR) is used during training, but it switches back to Token Choice (TC) for evaluation. The authors also admit that "Token rounding is not a token-choice routing method which creates difficulty for autoregressive inference." This is a major concern. The authors do not clearly explain the root cause of this 'difficulty'.
>
> > Furthermore, this training-inference mismatch constitutes a theoretical weakness. Although the results in Table 1 show acceptable PPL and Avg accuracy, it is unclear if this mismatch could negatively impact other aspects of the model, such as generation diversity or the prevalence of 'hallucinations'.
>
>
> We appreciate the reviewer raising this important point. We address the technical details and share some encouraging findings:
>
> **Inference:** Token rounding requires token sorting for each expert (Algorithm 4, step 4) similar to expert-choice routing, which breaks the causality required for autoregressive generation. Although such sorting is feasible during traing, inference requires processing tokens sequentially without future token leakage. This is why we use token rounding for training and stick with token-choice for inference.
>
> We acknowledge the mentioned weakness for token rounding, but argue that **token rounding provides a better pareto-frontier between inference quality and training efficiency** (10-20% higher TFLOPs than TC top-K, refer to Figure 7). We note that TR achieves much better inference quality compared to even EC adapted with a direct adoption of TC top-K during inference. To illustrate this, we provide benchmark results for expert-choice (EC) routing directly evaluated via token-choice top-K router. We also consider Mixture-of-Depth's approach [1] that trains an auxiliary router to predict the EC router's selection during validation. This baseline is referred as ''EC (aux router)''. We also adapt EC routing to TC routing by fine-tuning a learned TC top-K router on top of a trained EC router. This is the ''EC (ft TC router)'' baseline. Please find the evaluation results for MoEs of 2 different configurations in the tables provided below:

---

> ### Author Response · Authors · 2025-11-22
> **Response to W1, W2 continued**
>
> ### **0.5B params, 40B tokens, 2/64 activated**
>
> | Method            | Train  PPL   | Val  PPL     | Wino     | SIQA     | SciQ     | PIQA     | OBQA     | HS       | COPA     | CSQA     | BoolQ    | ArcE     | ArcC     | Avg      |
> | ----------------- | --------- | --------- | -------- | -------- | -------- | -------- | -------- | -------- | -------- | -------- | -------- | -------- | -------- | -------- |
> | **TR**            | **16.22** | **15.92** |  **51.4**    | 41.6     | 78.4     | **65.4** | 31.6     | **38.1** | 65.0     | 31.0     | **61.1**   | **57.4** | 29.1 | **50.0** |
> | EC                | 16.83     | 18.61     | 49.6     | 41.4     | 79.1     | 64.4     | **33.4** | 36.9     | 62.0     | **32.8** | 60.2     | 55.8     | 29.1 | 49.5     |
> | EC (aux router)   | 16.80     | 21.80     | 50.0     | 40.9     | 75.2     | 63.7     | 28.2     | 35.2     | 61.0     | 31.5     | 57.2     | 53.3     | 24.7     | 47.4     |
> | EC (ft TC router) | 16.81     | 16.98     | 50.0     |  41.7     | **79.7** | 64.9     | 31.6     | 36.8     | 63.0     | 32.1     | 60.7     | 54.6     | 27.4     | 49.3     |
> | TC top-K | 16.34 | 15.94 | 51.0 | **41.9** | 78.5 | 64.8 | 33.0 | **38.1** | **67.0** | 30.8 | 54.7 | 55.8 | **30.1** | 49.6 |
>
> ### **1.4B params, 100B tokens, 2/128 activated**
>
> | Method            | Train   PPL  | Val   PPL    | Wino     | SIQA     | SciQ     | PIQA     | OBQA     | HS       | COPA     | CSQA     | BoolQ    | ArcE     | ArcC     | Avg      |
> | ----------------- | --------- | --------- | -------- | -------- | -------- | -------- | -------- | -------- | -------- | -------- | -------- | -------- | -------- | -------- |
> | **TR**            | **13.31** | **13.22** | **52.8** | 41.8     | 80.8     | **68.7** | 33.0     | **43.4** | **67.0** | 33.6     | **60.2** | 60.7     | **29.8** | **52.0** |
> | EC                | 14.08     | 24.79     | 51.5     | 41.7     | 81.0     | 66.1     | 33.2  | 40.6     | 64.0     | 34.0     | 56.3     | 56.5     | 27.4     | 50.2     |
> | EC (aux router)   | 14.01     | 37.52     | 49.7     | 40.2     | 73.6     | 57.5     | 27.6     | 33.2     | 61.0     | 27.8     | 58.8     | 45.2     | 24.2     | 45.3     |
> | EC (ft TC router) | 14.24     | 14.75     | 52.2     | **42.6**     | 79.4     | 65.7     | 32.8     | 40.8     | 64.0     | 34.9     | 58.3     | 57.2     | 27.1     | 50.5     |
> | TC top-K  | 13.50 | 13.32 | 51.3 | 42.0 | **83.2** | 68.2 | **34.0** | **43.4** | 66.0 | **35.4** | 57.9 | **61.6** | 29.4 | **52.0**
>
>
> The training-inference discrepancy between TR and TC top-K is smaller than the discrepancy between EC and TC, even after TC fine-tuning. Note that **we do not apply any adaptation to TR and already observe robust performance during inference with token choice routing**.
>
> We acknowledge that the potential discrepancy might only be detectable on larger scale, but we argue that we do not find any significant performance degradation on 1-2B scale MoE with robustness ablations on $M_\mathrm{tile}$ (see our W2 response to reviewer LFjW). So we believe train-infer discrepancy of TR is a minor concern based on our new results.

---

> ### Author Response · Authors · 2025-11-22
> **Response to reviewer tULt (2/2)**
>
> ## W3 - loss convergence of TR
>
> > Moreover, the paper does not discuss the differences in loss convergence between the two methods (TR vs. TC) during the training phase.
>
> Thanks for this suggestion! We have plotted the training perplexity (100-step running average) of (1) 0.5B params, 40B tokens, 2/64 activated (2) 1.4B params, 100B tokens, 2/128 activated MoE in this anonymous link. We can observe that TR converge similarly as TC, with loss curve effectively overlaps and slightly lower at the end of training.
>
> [Training Perplexity of TR vs TC](https://anonymous.4open.science/r/8082-D1F0/training_perplexity.png)
>
> We will add this figure and the corresponding analysis to our paper in the next revision! Thanks for this suggestion!
>
> ## W4 - clarification of novelty
>
> > The paper highlights 'Ping-Pong' scheduling and asynchronous TMA as key to its IO-aware kernel design. While their application in SNaX is effective, these scheduling techniques themselves are not new. The authors should more clearly articulate that their core contribution lies in the specific application and adaptation of these techniques to the heavy-epilogue nature of MoE kernels, rather than claiming novelty for the techniques themselves.
>
>
> Thanks for the suggestion! **We would like to clarify that we do *not* claim we invent Ping-Pong scheduling, as Ping-Pong scheduling is already known in other places such as Flash Attention 3, but the application of Ping-Pong scheduling to address the increasing IO costs of fine-grained MoE kernel design is novel.** This design is missing from all current MoE training kernel designs which we will add a detailed discussion in the next revision of our paper.
>
> In Section 2, we have discussed why granularity will reduce TFLOPs despite iso-params and iso-FLOPS, and Ping-Pong scheduling is one ingradient for mitigating the throughput degradation from the increase of IO costs. We apply Ping-Pong primarily on down-proj forward and backward activation grad that has heavy epilogue. Before Hopper GPUs, such heavily epilogues would cause a severe slowdown but starting with Hopper's Ping-Pong scheduling we can have kernels that maintain high throughput in settings with heavy epilogues. So such application of overlapping IO with MMA compute for fine-grained MoEs becomes increasingly important.
>
>
>
> ## W5 - clarification of technical terms
>
> > The paper fails to clarify certain technical terms. For instance, what is the "state-of-the-art BF16 MoE kernel" that SNaX is compared against? Additionally, what does the "K dimension" (mentioned in Section 2.1) refer to in the context of GEMM?
>
>
> We apologize for the confusion. The "state-of-the-art BF16 MoE kernel" in our abstract is ScatterMoE. We use ScatterMoE to train a 7B MoE with FSDP-2 on a production-grade LLM training engine (which provides other training kernels such as Flash Attention 2, chunked cross entropy loss, etc.). We find that our SNaX kernel achieves 213 billion tokens a day with 64 H100 GPUs comparable to ScatterMoE's 225 billion tokens a day with 96 GPUs.
>
> In this paper, we follow standard GEMM notations in BLAS[1]: we have $A \in \mathbb{R}^{\mathbf{M}\times \mathbf{K}}$, $B \in \mathbb{R}^{\mathbf{K}\times \mathbf{N}}$, $C \in \mathbb{R}^{\mathbf{M}\times \mathbf{N}}$ for $C = A B$ with problem shape ($\mathbf{M},\mathbf{N},\mathbf{K}$). This notation is adopted by CUTLASS[2] which implements efficient GEMM on CUDA compatible devices. So we refer "K dimension in GEMM" as the dimension over which we are contracting/accumulating the tensor.
>
> Thanks for this suggestion! We will add this clarification in our next revision of the paper.
>
> Reference:
>
> [1] BLAS (Basic Linear Algebra Subprograms) https://www.netlib.org/blas/
>
> [2] Efficient GEMM in CUDA. NVIDIA CUTLASS GEMM, https://docs.nvidia.com/cutlass/media/docs/cpp/efficient_gemm.html

---

> > ### Author Response · Authors · 2025-12-03
> > **Response to Q1**
> >
> > ## Q1 - Potential numerical inaccuracy with mixed precision (BF16)
> >
> > Thanks for raising this question! We want to clarify that our down-proj epilogue will compute $dZ$, $Y_1$ (forward SwiGLU output), and $dS$ on registers with FP32 precision. We only save the final results of $dZ$ and $dS$ on BF16, so the rounding error only happens on the final HBM store step which is the same for any BF16 kernels. **Although SNaX is optimized for throughput, we are still careful on numerical accuracy. For example, we always keep router score $S$ in FP32 precision, and all on-register operations (e.g. SwiGLU/dSwiGLU activation, accumulation) are always conducted with FP32 precision.**
> >
> > This is an example relative numerical errors of SNaX compared to a reference FP32 MoE implemented on PyTorch. We initialize $W_1$ and $W_2$ from $\mathcal{N}{(0, 0.02)}$, and $X, dO \in \mathcal{N}{(0, 0.2)}$. **We can find that SNaX only occurs relative errors ~2% (<4 ULP of BF16) for BF16 and ~0.2% (<4 ULP of FP16) for FP16 on average.**
> >
> >
> > SNaX BF16, 7B MoE (microbatch size $T$ = 24576, embedding dim $d$ = 1536, expert intermediate size $n$ = 256, number of experts $E$ = 128, number of activated experts $K$ = 8)
> >
> > | |  mean absolute value | max absolute error |  mean relative error |
> > | --- | --- | --- |  --- |
> > | Forward output $O$ | 1.12e-3 |  6.1e-5  | 1.40%  |
> > | Backward activation gradient $dX$ | 1.60e-3 | 1.2e-4 | 1.82% |
> > | Backward weight gradient $dW2$ | 9.71e-3 |  4.9e-4 | 1.75% |
> > | Backward weight gradient $dW1$ | 9.71e-3 | 4.9e-4 | 1.42% |
> > | Backward weight gradient for router | 2.27e-2 | 4.9e-4 | 1.98% |
> >
> >
> > SNaX FP16, 7B MoE ($T$ = 24576, $d$ = 1536, $n$ = 256, $E$ = 128, $K$ = 8)
> >
> > | |  mean absolute value | max absolute error |  mean relative error |
> > | --- | --- | --- |  --- |
> > | Forward output $O$ | 1.12e-3 |  8e-6  | 0.174%  |
> > | Backward activation gradient $dX$ | 1.59e-3 | 1.5e-5 | 0.225% |
> > | Backward weight gradient $dW2$ | 9.70e-3 | 6.1e-5 | 0.217% |
> > | Backward weight gradient $dW1$ | 9.71e-3 | 6.1e-5 | 0.178% |
> > | Backward weight gradient for router | 2.27e-2 | 6.1e-5 | 0.255% |
> >
> >
> > We also verified the numerical accuracy of SNaX in a real 7B training run, where we do not observe a noticeable loss diff ($<1$\%) on the first 1000 steps. **All of our experiments are conducted on SNaX kernels and we do not find any training loss spikes or divergence.**

---

### Official Review · Reviewer_LFjW · 2025-11-01

**Soundness:** 3
**Presentation:** 3
**Contribution:** 3
**Rating:** 6
**Confidence:** 4

**Summary:**

This work presents a new efficient implementation of MoE operator for reducing the peaking GPU memory of MoE's training process. This work also introduces a grouped-GEMM–aware token rounding method for tackling the wasted tile computation, as the required computation is lower than the minimum value that a hardware unit can compute in one pass. This phenomenon often occurs when training the highly sparse MoEs since each expert is small and some of them may only receive a few tokens in one pass. This work conducts experiments on a broad scale of MoEs (1.4B-120B), and comparisons against ScatterMoE, MoMoE, and MegaBlocks demonstrate its effectiveness.

**Strengths:**

The writing is clear, and this work targets the efficiency bottlenecks of training highly sparse MoEs, which is an important research direction for the large models community. More and more recent LLMs and VLMs adopt the MoE structure, and as the trending analysis proposed in the appendix shows, recent MoEs tend to increase the sparsity of activated experts.

**Weaknesses:**

Weakness:
1. Although the motivation of token rounding aims to reduce tile padding waste in grouped GEMM, the proposed method is similar to load balancing and token dispatching strategies. There are many prior works on token-choice and expert-choice. The proposed method's novelty is limited. The related work 2.4 briefly discusses the routing method, but it's better to discuss the hybrid methods for tackling imbalance issues in detail.
2. May token rounding introduce hardware sensitivity? The strategy depends on grouped GEMM tile sizes and implementation details (which vary across GPUs/libraries/precisions). This could lead to unexpected behavior when open-sourcing the MoE models since the user may fine-tune it on different devices.
3. The risk of training–inference consistency and behavioral bias. This work trains with token rounding but evaluates with token-choice. Recent work suggests routing consistency matters in RL/align phases; train–infer routing mismatch may increase the chance of mode collapse and bias amplification.

**Questions:**

see weakness

---

> ### Author Response · Authors · 2025-11-22
> **Response to reviewer LFjW (1/3)**
>
> We thank reviewer LFjW for appreciation on our clear writing and a valuable target on the efficiency bottlenecks of training highly sparse MoEs. Below we will address reviewer LFjW's questions and concerns.
>
> ## W1 - novelty of token rounding
>
> > Although the motivation of token rounding aims to reduce tile padding waste in grouped GEMM, the proposed method is similar to load balancing and token dispatching strategies. There are many prior works on token-choice and expert-choice. The proposed method's novelty is limited. The related work 2.4 briefly discusses the routing method, but it's better to discuss the hybrid methods for tackling imbalance issues in detail.
>
> We appreciate the reviewer's observation that our goal of reducing grouped-GEMM padding is related to load-balancing and routing methods. However, **token rounding is orthogonal to existing routing / dispatching strategies: it solves a *tile quantization* problem rather than a load-balancing problem.** Existing works that drop or reroute tokens primarily target imbalanced *compute* across experts / devices, not the tile-aligned utilization of grouped GEMM.
>
> - **Routers that control expert loads via dropping / rerouting.**
>
> Switch Transformer [4] introduces an "expert capacity" factor: when the number of tokens routed to an expert exceeds a fixed capacity, some tokens (often with lowest scores) are skipped, or "dropped". DeepSeek-V2 [5] adopts token dropping during training, and He et al. [3] extend capacity-aware dropping / rerouting to inference to mitigate stragglers. All of these methods treat the capacity as a static per-expert limit. In contrast, **padding waste in grouped GEMM is inherently dynamic: it depends on the realized per-expert counts and the hardware tile size, and cannot be fixed by a single global capacity chosen before training.**
>
> Moreover, **dropping is not a good surrogate for token rounding.** If we modify our rounding procedure to always discard tokens in the first TC sorting step (equivalent to token dropping with capacity equal to the largest multiple of $M_\mathrm{tile}$ not exceeding the number of routed tokens for each expert in TC top-K routing), we observe a non-trivial degradation in downstream accuracy:
>
>
>
> ### **1.8B params, 40B tokens, 8/256 activated**
>
> | Method            | Val  PPL     | Wino     | SIQA     | SciQ     | PIQA     | OBQA     | HS       | COPA     | CSQA     | BoolQ    | ArcE     | ArcC     | Avg      |
> | ----------------- | --------- | --------- | -------- | -------- | -------- | -------- | -------- | -------- | -------- | -------- | -------- | -------- | -------- |
> | **TR**            | **13.10** | 53.4 |  **42.1**   |  81.7   |  **69.6**  |  **35.2** |  **45.3**   |  **70.0** |  33.2   |  **61.4**    |  63.0  |  **33.4**  |  **53.5** |
> | Token dropping to last $M_\mathrm{tile}$-multiples | 13.19  | **55.4** | 41.6 |  **82.2** | 68.6 | 34.8 | 45.0  | 69.0  | **34.0** | 54.4 |  **63.5** | 31.4 | 52.7 |
>
> ---
>
> ### **1.4B params, 100B tokens, 2/128 activated**
>
> | Method            | Val  PPL     | Wino     | SIQA     | SciQ     | PIQA     | OBQA     | HS       | COPA     | CSQA     | BoolQ    | ArcE     | ArcC     | Avg      |
> | ----------------- | --------- | --------- | -------- | -------- | -------- | -------- | -------- | -------- | -------- | -------- | -------- | -------- | -------- |
> | **TR**            | **13.22** | **52.8** | 41.8 | 80.8 | **68.7** | **33.0** | **43.4** | **67.0** | 33.6 | **60.2** | **60.7** | 29.8 | **52.0** |
> | Token dropping to last $M_\mathrm{tile}$-multiples | 13.29  | 50.0 | **42.2** | **81.7** | 68.3 | 31.2 | 43.3 | 66.0 | **34.3** | 56.6 | 59.5 | **30.8**  | 51.3 |
>
> ---
>
> To our knowledge, the closest router-side work to ours is Rectify-Router [2], which also focuses on expert parallelism. Rectify-Router defines expert capacities and then (i) reroutes overflow tokens to other experts on the same GPU and (ii) replaces padding of under-loaded experts with high-scoring tokens. **Crucially, Rectify-Router still ignores grouped-GEMM tile sizes: it tries to fill leftover capacity but does not align the final expert counts to the GEMM tile, and thus does not directly minimize tile padding FLOPs.**

---

> > ### Author Response · Authors · 2025-11-22
> > **Response to W1 continued**
> >
> > - **MoE / grouped-GEMM methods that reduce padding waste.**
> >
> > On the systems side, the closest prior work is TMA-adaptive FP8 Grouped GEMM [1], which eliminates pre-padding by dynamically managing a pool of TMA descriptors. Their primary focus is to reduce *excessive GMEM traffic* caused by static descriptors; they do not change the token–expert assignment and therefore do not control the *FLOPs* wasted by non–tile-aligned group sizes. Our method is complementary: **we adjust the expert token counts themselves via a constrained rounding scheme, and our grouped-GEMM kernel then updates TMA descriptors online to exploit these tile-aligned sizes.**
> >
> > We thank the reviewer for bringing up this novelty discussion. We will add this discussion in the next revision of paper.
> >
> >
> > Reference:
> >
> > [1] Su, Zhongling, et al. "TMA-Adaptive FP8 Grouped GEMM: Eliminating Padding Requirements in Low-Precision Training and Inference on Hopper." arXiv:2508.16584.
> >
> > [2] Zeng, Zhiyuan, et al. "Turn Waste into Worth: Rectifying Top-k Router of MoE." EMNLP 2024.
> >
> > [3] He, Shwai, et al. "Capacity-Aware Inference: Mitigating the Straggler Effect in Mixture of Experts." arXiv:2503.05066.
> >
> > [4] Fedus, William, Barret Zoph, and Noam Shazeer. "Switch transformers: Scaling to trillion parameter models with simple and efficient sparsity." JMLR 2022.
> >
> > [5] Liu, Aixin, et al. "Deepseek-v2: A strong, economical, and efficient mixture-of-experts language model." arXiv:2405.04434.

---

> ### Author Response · Authors · 2025-11-22
> **Response to reviewer LFjW (2/3)**
>
> ## W2 - hardware sensitivity
>
> > May token rounding introduce hardware sensitivity? The strategy depends on grouped GEMM tile sizes and implementation details (which vary across GPUs/libraries/precisions). This could lead to unexpected behavior when open-sourcing the MoE models since the user may fine-tune it on different devices.
>
> Thanks for providing this insightful suggestions! We acknowledge that the hardware mismatches can cause potential discrepancy between pretraining and fine-tuning. In response, we ablate the robustness of tile size $M_\mathrm{tile}$ on TR under the settings of sparse MoE training, and we can also find TR is robust to tile sizes within reasonable ranges. In particular, we often observe minimal quality degradation on downstream tasks until average number of routed tokens per expert is about the same as $M_\mathrm{tile}$. We also note that $M_\mathrm{tile}$ = 512 in the following table is more than the largest permissible tile size in both NVidia Hopper and Blackwell GPUs (often 256 in both cases).
>
>
> ### **0.5B params, 40B tokens, 2/64 activated** (avg tokens per expert = 512)
>
> | Method                | Val PPL       | Wino     | SIQA     | SciQ     | PIQA     | OBQA     | HS       | COPA     | CSQA     | BoolQ    | ArcE     | ArcC     | Avg      |
> | --------------------- | --------- | -------- | -------- | -------- | -------- | -------- | -------- | -------- | -------- | -------- | -------- | -------- | -------- |
> | TR ($M_\text{tile}=64$)      | **15.90** | 51.3     | 41.7 | 78.1     | 65.6 | 31.4     | 37.9     | **67.0** | **32.4** | 59.8     | 57.2     | 28.8     | 50.1     |
> | TR ($M_\text{tile}=128$) | 15.92     | 51.4     | 41.6     | 78.4     | 65.4     | 31.6     | **38.1** | 65.0     | 31.0     | **61.1** | 57.4     | 29.1     | 50.0     |
> | TR ($M_\text{tile}=256$)     | 16.00     | 51.7     | 41.4     | 78.7     | **66.3**     | 32.4 | 37.7     | **67.0** | 31.3     | 60.1     | **58.2** | 29.1     | **50.4** |
> | TR ($M_\text{tile}=512$)     | 16.17     | **52.5** | 41.2     | **80.2** | 65.2     | 32.0     | 37.9     | 62.0     | 31.0     | 59.4     | 57.2     | **30.4** | 49.9     |
> | TC top-K            | 15.94     | 51.0     | **41.9**     | 78.5     | 64.8     | **33.0**     | **38.1**     | **67.0**     | 30.8     | 54.7     | 55.8     | 30.1     | 49.6     |
>
> ---
>
> ### **1.8B params, 40B tokens, 8/256 activated** (avg tokens per expert = 512)
>
> | Method                | PPL       | Wino     | SIQA     | SciQ     | PIQA     | OBQA     | HS       | COPA     | CSQA     | BoolQ    | ArcE     | ArcC     | Avg      |
> | --------------------- | --------- | -------- | -------- | -------- | -------- | -------- | -------- | -------- | -------- | -------- | -------- | -------- | -------- |
> | TR ($M_\text{tile}=64$)      | **13.07** | 52.3     | **42.9** | **82.7** | 69.4     | **35.4** | **45.6** | **70.0** | 32.4     | 56.6     | 64.4     | 31.4     | 53.0     |
> | TR ($M_\text{tile}=128$) | 13.10     | **53.4** | 42.1     | 81.7     | **69.6** | 35.2     | 45.3     | **70.0** | 33.2     | **61.4** | 63.0     | **33.4** | **53.5** |
> | TR ($M_\text{tile}=256$)     | 13.13     | 52.0     | 41.6     | 82.1     | 69.2     | **35.4** | 45.3     | 69.0     | **34.2** | 58.0     | **65.6** | 32.1     | 53.1     |
> | TR ($M_\text{tile}=512$)     | 13.56     | 53.0     | 41.8     | 81.2     | 68.4     | 34.0     | 44.2     | 68.0     | 33.3     | 58.1     | 59.5     | 30.1     | 52.0     |
> | TC top-K            | 13.12     | 50.1     | 42.9     | 81.3     | 69.8     | 33.8     | 45.2     | 71.0     | 34.1     | 56.7     | 64.6     | 31.1     | 52.8     |
>
>
> ---
>
>
> **On the quality side, $M_\mathrm{tile}$ can be treated as a flexible hyperparameter to balance the quality and speed. If the user wants a smaller potential discrepancy during fine-tuning and do not focus on the speed, the user could always choose a smaller tile size and with tile size 1 recovers the exact TC top-$K$ routing.** Since during end-user finetuning, datasets are typically smaller in size, its reasonable to train with tile sizes that might not be hardware friendly. **And that's why we emphasize more on the training efficiency during pretraining instead of fine-tuning.**

---

> ### Author Response · Authors · 2025-11-22
> **Response to reviewer LFjW (3/3)**
>
> ## W3 - training-inference consistency
>
> > The risk of training–inference consistency and behavioral bias. This work trains with token rounding but evaluates with token-choice. Recent work suggests routing consistency matters in RL/align phases; train–infer routing mismatch may increase the chance of mode collapse and bias amplification.
>
>
>
> We appreciate the reviewer raising this important point. We acknowledge the mentioned weakness for token rounding, but argue that **token rounding provides a better pareto-frontier between inference quality and training efficiency** (10-20% higher TFLOPs than TC top-K, refer to Figure 7). It should also be noted that TR also achieves much better inference quality compared to even EC adapted with a direct adoption of TC top-K during inference.
>
> To illustrate this, we provide benchmark results for expert-choice (EC) routing directly evaluated via token-choice top-K router. We also consider Mixture-of-Depth's approach [1] that trains an auxiliary router to predict the EC router's selection during validation. This baseline is referred as ''EC (aux router)''. We also adapt EC routing to TC routing by fine-tuning a learned TC top-K router on top of a trained EC router. This is the ''EC (ft TC router)'' baseline.
>
> Please find the evaluation results for MoEs of 2 different configurations in the tables provided below:
>
> ### **0.5B params, 40B tokens, 2/64 activated**
>
> | Method            | Train  PPL   | Val  PPL     | Wino     | SIQA     | SciQ     | PIQA     | OBQA     | HS       | COPA     | CSQA     | BoolQ    | ArcE     | ArcC     | Avg      |
> | ----------------- | --------- | --------- | -------- | -------- | -------- | -------- | -------- | -------- | -------- | -------- | -------- | -------- | -------- | -------- |
> | **TR**            | **16.22** | **15.92** |  **51.4**    | 41.6     | 78.4     | **65.4** | 31.6     | **38.1** | 65.0     | 31.0     | **61.1**   | **57.4** | 29.1 | **50.0** |
> | EC                | 16.83     | 18.61     | 49.6     | 41.4     | 79.1     | 64.4     | **33.4** | 36.9     | 62.0     | **32.8** | 60.2     | 55.8     | 29.1 | 49.5     |
> | EC (aux router)   | 16.80     | 21.80     | 50.0     | 40.9     | 75.2     | 63.7     | 28.2     | 35.2     | 61.0     | 31.5     | 57.2     | 53.3     | 24.7     | 47.4     |
> | EC (ft TC router) | 16.81     | 16.98     | 50.0     |  41.7     | **79.7** | 64.9     | 31.6     | 36.8     | 63.0     | 32.1     | 60.7     | 54.6     | 27.4     | 49.3     |
> | TC top-K | 16.34 | 15.94 | 51.0 | **41.9** | 78.5 | 64.8 | 33.0 | **38.1** | **67.0** | 30.8 | 54.7 | 55.8 | **30.1** | 49.6 |
>
> ### **1.4B params, 100B tokens, 2/128 activated**
>
> | Method            | Train   PPL  | Val   PPL    | Wino     | SIQA     | SciQ     | PIQA     | OBQA     | HS       | COPA     | CSQA     | BoolQ    | ArcE     | ArcC     | Avg      |
> | ----------------- | --------- | --------- | -------- | -------- | -------- | -------- | -------- | -------- | -------- | -------- | -------- | -------- | -------- | -------- |
> | **TR**            | **13.31** | **13.22** | **52.8** | 41.8     | 80.8     | **68.7** | 33.0     | **43.4** | **67.0** | 33.6     | **60.2** | 60.7     | **29.8** | **52.0** |
> | EC                | 14.08     | 24.79     | 51.5     | 41.7     | 81.0     | 66.1     | 33.2  | 40.6     | 64.0     | 34.0     | 56.3     | 56.5     | 27.4     | 50.2     |
> | EC (aux router)   | 14.01     | 37.52     | 49.7     | 40.2     | 73.6     | 57.5     | 27.6     | 33.2     | 61.0     | 27.8     | 58.8     | 45.2     | 24.2     | 45.3     |
> | EC (ft TC router) | 14.24     | 14.75     | 52.2     | **42.6**     | 79.4     | 65.7     | 32.8     | 40.8     | 64.0     | 34.9     | 58.3     | 57.2     | 27.1     | 50.5     |
> | TC top-K  | 13.50 | 13.32 | 51.3 | 42.0 | **83.2** | 68.2 | **34.0** | **43.4** | 66.0 | **35.4** | 57.9 | **61.6** | 29.4 | **52.0**
>
> The training-inference discrepancy between TR and TC top-K is smaller than the discrepancy between EC and TC, even after TC fine-tuning. Note that **we do not apply any adaptation to TR and already observe robust performance during inference with token choice routing**.
>
> We acknowledge that the potential bias amplication might only be detectable on larger scale, but we argue that we do not find any significant performance degradation on 1-2B scale MoE with robustness ablations on $M_\mathrm{tile}$. TR is already outperforms the EC and token dropping (see W1) approach for train-infer discrepancy, and both approaches are still employed in large-scale pretraining. So we believe train-infer discrepancy of TR is a minor concern based on our new results.
>
>
> **Reference**:
>
> [1] Raposo, David, et al. "Mixture-of-depths: Dynamically allocating compute in transformer-based language models."

---

### Official Review · Reviewer_fRZc · 2025-11-12

**Soundness:** 3
**Presentation:** 3
**Contribution:** 3
**Rating:** 6
**Confidence:** 4

**Summary:**

This paper introduces SNaX, a co-design solution for training fine-grained and sparse MoE models efficiently. The authors identify that increasing MoE granularity and sparsity shifts computation from compute-bound to memory-bound. They propose three contributions: (1) a memory-efficient backward pass maintaining constant activation memory regardless of expert granularity (45% reduction), (2) IO-aware kernels with asynchronous operations achieving 1.87x throughput improvement, and (3) token rounding that aligns token counts to GPU tile sizes, reducing wasted computation by up to 18%.

**Strengths:**

The paper correctly identifies that modern MoE architectures face a critical shift from compute-bound to memory-bound operations as granularity and sparsity increase, supported by compelling arithmetic intensity analysis. The memory-efficient backward pass cleverly avoids materializing Y₂ through computation reordering, achieving 45% memory reduction while maintaining mathematical equivalence. Token rounding addresses the overlooked tile quantization problem where padding wastes significant compute in sparse MoEs, yielding 18% additional speedup with maintained accuracy. Comprehensive experiments across 1.4B-120B scales show consistent improvements over ScatterMoE (1.87x throughput) and MoMoE, with the planned open-source release providing valuable production-quality kernels to the community.

**Weaknesses:**

The fundamental incompatibility between token rounding (training) and token-choice (inference) is acknowledged but poorly explained, with authors only stating it "creates difficulty" without technical details or solutions. The training-inference mismatch may impact generation quality beyond perplexity (diversity, hallucinations) but remains uninvestigated. Heavy Hopper-specific dependence (TMA, tile sizes) limits portability with no analysis on A100/TPU performance degradation. While presenting ping-pong scheduling prominently, these are known techniques - the contribution is adaptation to MoE epilogues, not algorithmic novelty. Critical details like the "state-of-the-art BF16 MoE kernel" baseline and GEMM "K dimension" remain undefined, hindering reproducibility.

**Questions:**

The paper focuses on Hopper GPU optimizations (TMA, async operations). What is the performance degradation on A100 or V100 GPUs? Is SNaX still beneficial without these hardware features, or does it become worse than baselines?

---

> ### Author Response · Authors · 2025-11-21
> **Response to reviewer fRZc (1/4)**
>
> We thank reviewer fRZc for the valuable feedback and appreciation of our memory-efficient backward implementation, consistent improvements over ScatterMoE and other baselines, and the analysis on the overlooked tile quantization issue for sparse MoEs. Please see the following clarification.
>
> ## W1 - incompatibility between token rounding (training) and token-choice (inference)
>
> > The fundamental incompatibility between token rounding (training) and token-choice (inference) is acknowledged but poorly explained, with authors only stating it "creates difficulty" without technical details or solutions.
>
> We appreciate the reviewer raising this important point. We address the technical details and share some encouraging findings:
>
> **Inference:** Token rounding requires token sorting for each expert (Algorithm 4, step 4) similar to expert-choice routing, which breaks the causality required for autoregressive generation. Although such sorting is feasible during traing, inference requires processing tokens sequentially without future token leakage.
>
> We acknowledge the mentioned weakness for token rounding, but argue that **token rounding provides a better pareto-frontier between inference quality and training efficiency** (10-20% higher TFLOPs than TC top-K, refer to Figure 7). It should also be noted that TR also achieves much better inference quality compared to even EC adapted with a direct adoption of TC top-K during inference. To illustrate this, we provide benchmark results for expert-choice (EC) routing directly evaluated via token-choice top-K router. We also consider Mixture-of-Depth's approach [1] that trains an auxiliary router to predict the EC router's selection during validation. This baseline is referred as ''EC (aux router)''. We also adapt EC routing to TC routing by fine-tuning a learned TC top-K router on top of a trained EC router. This is the ''EC (ft TC router)'' baseline. Please find the evaluation results for MoEs of 2 different configurations in the tables provided below:
>
> ### **0.5B params, 40B tokens, 2/64 activated**
>
> | Method            | Train  PPL   | Val  PPL     | Wino     | SIQA     | SciQ     | PIQA     | OBQA     | HS       | COPA     | CSQA     | BoolQ    | ArcE     | ArcC     | Avg      |
> | ----------------- | --------- | --------- | -------- | -------- | -------- | -------- | -------- | -------- | -------- | -------- | -------- | -------- | -------- | -------- |
> | **TR**            | **16.22** | **15.92** |  **51.4**    | 41.6     | 78.4     | **65.4** | 31.6     | **38.1** | 65.0     | 31.0     | **61.1**   | **57.4** | 29.1 | **50.0** |
> | EC                | 16.83     | 18.61     | 49.6     | 41.4     | 79.1     | 64.4     | **33.4** | 36.9     | 62.0     | **32.8** | 60.2     | 55.8     | 29.1 | 49.5     |
> | EC (aux router)   | 16.80     | 21.80     | 50.0     | 40.9     | 75.2     | 63.7     | 28.2     | 35.2     | 61.0     | 31.5     | 57.2     | 53.3     | 24.7     | 47.4     |
> | EC (ft TC router) | 16.81     | 16.98     | 50.0     |  41.7     | **79.7** | 64.9     | 31.6     | 36.8     | 63.0     | 32.1     | 60.7     | 54.6     | 27.4     | 49.3     |
> | TC top-K | 16.34 | 15.94 | 51.0 | **41.9** | 78.5 | 64.8 | 33.0 | **38.1** | **67.0** | 30.8 | 54.7 | 55.8 | **30.1** | 49.6 |
>
> ### **1.4B params, 100B tokens, 2/128 activated**
>
> | Method            | Train   PPL  | Val   PPL    | Wino     | SIQA     | SciQ     | PIQA     | OBQA     | HS       | COPA     | CSQA     | BoolQ    | ArcE     | ArcC     | Avg      |
> | ----------------- | --------- | --------- | -------- | -------- | -------- | -------- | -------- | -------- | -------- | -------- | -------- | -------- | -------- | -------- |
> | **TR**            | **13.31** | **13.22** | **52.8** | 41.8     | 80.8     | **68.7** | 33.0     | **43.4** | **67.0** | 33.6     | **60.2** | 60.7     | **29.8** | **52.0** |
> | EC                | 14.08     | 24.79     | 51.5     | 41.7     | 81.0     | 66.1     | 33.2  | 40.6     | 64.0     | 34.0     | 56.3     | 56.5     | 27.4     | 50.2     |
> | EC (aux router)   | 14.01     | 37.52     | 49.7     | 40.2     | 73.6     | 57.5     | 27.6     | 33.2     | 61.0     | 27.8     | 58.8     | 45.2     | 24.2     | 45.3     |
> | EC (ft TC router) | 14.24     | 14.75     | 52.2     | **42.6**     | 79.4     | 65.7     | 32.8     | 40.8     | 64.0     | 34.9     | 58.3     | 57.2     | 27.1     | 50.5     |
> | TC top-K  | 13.50 | 13.32 | 51.3 | 42.0 | **83.2** | 68.2 | **34.0** | **43.4** | 66.0 | **35.4** | 57.9 | **61.6** | 29.4 | **52.0**
>
> The training-inference discrepancy between TR and TC top-K is smaller than the discrepancy between EC and TC, even after TC fine-tuning. Note that **we do not apply any adaptation to TR and already observe robust performance during inference with token choice routing**.
>
> **Reference**:
>
> [1] Raposo, David, et al. "Mixture-of-depths: Dynamically allocating compute in transformer-based language models."

---

> ### Author Response · Authors · 2025-11-21
> **Response to reviewer fRZc (2/4)**
>
> ## W2, Q1 - portability/transferability to other hardwares (V100, A100, TPU)
>
> > Heavy Hopper-specific dependence (TMA, tile sizes) limits portability with no analysis on A100/TPU performance degradation.
>
> > The paper focuses on Hopper GPU optimizations (TMA, async operations). What is the performance degradation on A100 or V100 GPUs? Is SNaX still beneficial without these hardware features, or does it become worse than baselines?
>
> The SNaX approach can be extended to older generations of hardware like NVIDIA A100s or V100s. However, since the older generations of GPUs are not actually asynchronous by nature, the kernel design for the older generation if GPUs is much more simpler. These older GPUs also don’t require a carefully designed scheduling, making it much easier to get peak performance.
>
> In particular,
> - **Activation memory efficiency of SNaX still holds regardless of GPU generation (and even for TPUs).**
> - **Improvement from router top-K, expert aggregation and an alternative computational path for $dS$, and heavy fusion still holds, but might have less impressive improvements over baselines such as ScatterMoE and MoMoE.** This is because older generation GPUs tend to have less peak TFLOPs/HBM bandwidth for the roofline and it is much easier to be compute-bound than memory bound.
> - We cannot apply techniques that overlap MMA with IO, so for the 2 Grouped GEMMs that have heavy IO (down-proj forward and backward activation grad computation), we can expect similar runtime.
>
> Therefore, we still have improvements over ScatterMoE and MoMoE but would less impressive compared to improvements achieved on Hopper and Blackwell in the future.
>
> **We aim to update our benchmark results for forward computation on Blackwell GPUs soon.** It also features overlapping IO with compute and more fine-grained control of each warp's role as the new UMMA (Unified Matrix Multiply-Accumulate) instruction and Tensor Memory (TMEM) alleviates the register pressure and allows dedicated warps for math and IO. **The more fine-grained controls and asynchrony allow for a much higher speedup from a carefully-designed kernel over a naive implementation.**
>
> We focus on Hopper & Blackwell because
>
> 1. **the problem of MoE low efficiency is more severe on newer hardware generation as the matmul gets faster relative to memory IO**. Older generations of GPU require less compute to reach the hardware peak, so the higher IO costs are of less concerns for them. However, newer generations of GPUs have much higher GEMM FLOPS and without SNaX, IO costs would overshadow the speedup from increasing GEMM FLOPS. This is why we focus on Hopper and Blackwell GPUs.
>
> 2. **the trend of accelerator design has moved towards more asynchrony (Hopper, Blackwell, AMD MI350 and MI400) creating new opportunity to overlap compute and memory IO.** Otherwise, synchronous IO costs would overshadow the boost from compute.
>
> 3. **these accelerators are now the majority of AI compute**

---

> > ### Author Response · Authors · 2025-11-21
> > **Response to reviewer fRZc (3/4)**
> >
> > ## W3 - Ping-Pong scheduling
> >
> > > While presenting ping-pong scheduling prominently, these are known techniques - the contribution is adaptation to MoE epilogues, not algorithmic novelty.
> >
> > We would like to clarify that we do *not* claim that we invent Ping-Pong scheduling. Ping-Pong scheduling is known in other places such as Flash Attention 3, but the application of Ping-Pong scheduling to address the increasing IO cost of fine-grained MoE kernel design is novel. In Section 2, we discuss why increasing granularity reduces MFU despite iso-params and iso-FLOPS, and how Ping-Pong scheduling can be used for mitigating the throughput degradation increased IO cost.
> >
> > We apply Ping-Pong primarily on down-proj forward and backward activation gradient computation since they have heavy epilogues. Before Hopper GPUs, such heavily epilogues caused a severe decrease in throughput but starting with Hopper's Ping-Pong we can have kernels that maintain high throughput. In Blackwell GPUs, we also have "Ping-Pong" in spirit that epilogue warps process results from Blackwell-specific memory unit (Tensor Memory) in 1 stage while MMA accumulation happens in another stage. **So such application of overlapping IO with MMA compute to address the IO costs of fine-grained MoEs is applicable beyond Hopper**.

---

> > > ### Author Response · Authors · 2025-11-21
> > > **Response to reviewer fRZc (4/4)**
> > >
> > > ## W4 - details of baseline
> > >
> > > > Critical details like the "state-of-the-art BF16 MoE kernel" baseline and GEMM "K dimension" remain undefined, hindering reproducibility.
> > >
> > > We are sorry for the confusion. The "state-of-the-art BF16 MoE kernel" in our abstract is ScatterMoE. We use ScatterMoE to train a 7B moe with FSDP-2 on a production-grade LLM training engine (with multiple other kernels such as Flash Attention 2, chunked cross entropy loss, etc.). We find that our SNaX kernel achieves 213 billion tokens a day with 64 GPUs comparable to ScatterMoE's 225 billion tokens a day with 96 GPUs.
> > >
> > > In this paper, we follow standard GEMM notations in BLAS[1]: we have $A \in \mathbb{R}^{\mathbf{M}\times \mathbf{K}}$, $B \in \mathbb{R}^{\mathbf{K}\times \mathbf{N}}$, $C \in \mathbb{R}^{\mathbf{M}\times \mathbf{N}}$ for $C = A B$ with problem shape ($\mathbf{M},\mathbf{N},\mathbf{K}$). This notation is adopted by CUTLASS[2] which implements efficient GEMM on CUDA. So we refer "K dimension in GEMM" as the dimension we are contracting/accumulating the tensor. A tiled matrix multiply will each time take a tile of ($M_\mathrm{tile}, N_\mathrm{tile}, K_\mathrm{tile}$) from ($\mathbf{M},\mathbf{N},\mathbf{K}$), and we parallelize over $\mathbf{M}$, $\mathbf{N}$, and accumulate over $\mathbf{K}$.
> > >
> > > **References**:
> > >
> > > [1] BLAS (Basic Linear Algebra Subprograms) https://www.netlib.org/blas/
> > >
> > > [2] Efficient GEMM in CUDA. NVIDIA CUTLASS GEMM, https://docs.nvidia.com/cutlass/media/docs/cpp/efficient_gemm.html

---

### Author Response · Authors · 2025-11-26
**Response to all reviewers**

We sincerely thank all reviewers for their constructive feedback. We are encouraged by the reviewers' shared recognition of our work on identifying and tackling the efficiency bottlenecks of fine-grained and sparse MoEs, consistently SOTA efficiency gains of SNaX's activation memory efficiency and training throughput and future impact for many researchers. Besides our novel contribution on the MoE kernel design, the successful application of our token rounding method to address tile quantization problem in sparse MoE training is also acknowledged by reviewer fRZc and tULt. Since submission, we have conducted extensive new experiments and analyses and updated the manuscript:

- As mentioned by the reviewers, our token rounding method (Sec 5) theoretically could lead to  training-inference inconsistency. This is not unlike other popular routing methods such as expert-choice (as far as we know, only token-choice routing would not have a training-inference inconsistency). **We view this as a tradeoff between efficiency and potential train-inference mismatch: token rounding gives most or all of the training efficiency of expert choice, while the train-inference mismatch is much smaller (close to negligible in our experiment). To validate this, we have added comparison with expert-choice routing with adaptions, and token dropping method applied to token-choice routing.** From our experiments, we observe that token rounding consistently achieves better downstream accuracy and validation perplexity than expert choice. The new results are included in Table 1. We also validate the robustness of token rounding w.r.t. tile sizes in Table 8 (page 18), and identify similar training convergence between TR and TC routing in Figure 10 (page 17).

- We have refined our experiments and technical content in the paper in accordance with the reviewers' feedback. We have added proper citations to Ping-Pong, discussed the related routing methods with token rounding, and provided detailed on-Figure annotations and in-text analyses on Figure 3 (Figure 6 in previous draft). Figure 3 and Section 4 now provide sufficient insights on the ingredients behind SNaX's speedup.

- **To further validate our method, we have added a comparison with the fastest MoE forward baseline we can find: DeepGEMM (originally by DeepSeek, recently rewritten and further optimized by NVidia) which provides highly optimized Grouped GEMM.** For an even stronger comparison, we further equip their Grouped GEMM with our optimized router, gather, SwiGLU, and expert aggregation kernels (we call this DeepGEMM++, we validate on NCU that all of these kernels can outperform Megatron, MoMoE, MegaBlocks related implementations) to assess the best possible forward throughput on top of the DeepGEMM library. **For a 30B MoE forward pass, SNaX still demonstrates a 1.11x - 1.43x faster and 1.22 - 2.44x more activation memory-efficient, and these results are consistently more prominent when expert granularity increases, as shown in Figure 6 in the revised manuscript.** Such strong results further validate the compute efficiency of SNaX and potential community impact for the AI research community.

Sincrely,

8082 Authors

---

### Meta-Review · Area_Chair_tLPK · 2026-01-04

**Summary:**

The reviewers’ concerns that informed the decision can be summarized as follows:

1. Reviewers were concerned that token rounding is used during training but token-choice routing is used at inference, potentially affecting generation quality and robustness.

2. Several reviewers questioned the heavy reliance on Hopper-specific features (e.g., TMA, asynchronous execution, tile sizes) and asked whether the method generalizes to A100/V100 or other accelerators.

3. Some reviewers noted that ping-pong scheduling and load-balancing–like routing ideas are known, and asked for clearer positioning of what is novel versus adapted.

4. Reviewers requested clarification of baselines (e.g., what constitutes the “state-of-the-art BF16 MoE kernel”) and more precise definitions of kernel parameters.

These concerns framed the initial borderline sentiment in some reviews.

**Reviewer Concerns:**

Concerns addressed by the rebuttal:

1. The authors provided extensive new experiments comparing token rounding with expert-choice routing, auxiliary-router variants, and fine-tuned token-choice routers. Results show that token rounding yields a better efficiency–quality Pareto frontier and that the train–inference discrepancy is smaller than for existing alternatives. This directly addresses the main conceptual concern.

2. The rebuttal clearly explains which benefits are hardware-agnostic (activation memory reduction, algorithmic reordering) and which are Hopper/Blackwell-specific (overlapping IO with compute). The authors reasonably justify focusing on newer accelerators where MoE inefficiency is most severe, while clarifying that SNaX does not regress on older GPUs.

3. The authors explicitly state that ping-pong scheduling itself is not new, but argue convincingly that its application to fine-grained MoE epilogues and memory-bound regimes is novel. Token rounding is clearly distinguished from load-balancing or token-dropping methods by its tile-quantization objective.

4. Ambiguities about baselines and GEMM notation were clarified, with ScatterMoE explicitly identified and kernel details aligned with standard BLAS/CUTLASS conventions.

Concerns still outstanding (but non-blocking):

1. While the discussion of non-Hopper hardware is convincing at a conceptual level, empirical results on non-NVIDIA accelerators are not provided. This is a reasonable limitation given the paper’s focus and does not undermine correctness.

2. Some reviewers may still view parts of the contribution as systems-level adaptation rather than new algorithms. However, this is appropriate for the paper’s primary area (infrastructure, kernels, systems).

Overall, no critical concerns remain unresolved.

**Reviewer Scores:**

Reviewer fRZc (initial rating: 6  marginally above threshold):
Likely increase to 8. The detailed responses on training–inference mismatch, portability, and baselines directly address their main weaknesses.

Reviewer LFjW (initial rating: 6 marginally above threshold):
Likely remain at 6 or increase to 8. Novelty concerns about token rounding are substantially clarified, and robustness analyses strengthen confidence.

Other reviewers (positive):
Likely unchanged or slightly increased, as the rebuttal reinforces already positive assessments with stronger evidence and clearer positioning.

---

### Decision · Program_Chairs · 2026-01-26

Accept (Poster)